# Comparing some aspects of the ocean hydrography, circulation and sea ice between NEMO3.6 LIM3 and LIM2

Petteri Uotila[1], Doroteaciro Iovino[2], Martin Vancoppenolle[3], Mikko Lensu[1], and Clement Rousset[3]

[1]Finnish Meteorological Institute, Helsinki, Finland
[2]Fondazione Centro Euro-Mediterraneo sui Cambiamenti Climatici (CMCC), Bologna, Italy
[3]Sorbonne Universités (UPMC Paris 6), LOCEAN-IPSL, CNRS/IRD/MNHN, Paris, France

*Correspondence to:* Petteri Uotila (petteri.uotila@fmi.fi)

**Abstract.** A set of hindcast simulations with the new NEMO3.6 ocean-ice model in the ORCA1 grid and forced by the DFS5.2 atmospheric data was performed from 1958–2012. Simulations differed in their sea-ice component: the old standard version LIM2 and its successor LIM3. Main differences between these sea-ice models are the parameterisations of sub-grid-scale sea-ice thickness distribution, ice deformation, thermodynamic processes, and sea-ice salinity. Our main objective was to diagnose the ocean-ice sensitivity to the updated NEMO-LIM3 sea-ice physics. Additional sensitivity simulations were carried out for the attribution of observed differences between the two main simulations.

In the Arctic, NEMO-LIM3 compares better with observations by realistically reproducing the sea-ice extent decline during the last few decades due to its multi-category sea-ice thickness. In the Antarctic, NEMO-LIM3 more realistically simulates the seasonal evolution of sea-ice extent than NEMO-LIM2. In terms of oceanic properties, improvements are not as evident, although NEMO-LIM3 reproduces a more realistic hydrography in the Labrador Sea and in the Arctic Ocean, including a reduced cold temperature bias of the Arctic intermediate water at 250 m. In the extra-polar regions, the oceanographic conditions of the two NEMO-LIM versions remain relatively similar, although they slowly drift apart over decades. This drift is probably associated with a more effective in deep water formation at the coastal waters of Antarctica manifested as deeper mixed layers.

## 1 Introduction

Sea ice is an important part of Earth's climate system because it effectively regulates the amount of energy being transferred between the atmosphere and oceans (Vaughan et al., 2013). Our current understanding on sea-ice related climate dynamics is incorporated in complex yet realistic climate models consisting of a sea-ice model component which is coupled to atmospheric and oceanic components (Griffies, 2004). In these models, sea ice can affect the ocean circulation and hydrography through ocean-ice interactions (see for example, Goosse and Fichefet, 1999; Kjellsson, 2015). To understand the effect of sea ice on the ocean, coupled global ocean-ice models, where the coupled atmospheric component is replaced with prescribed atmospheric forcing, can be used (Griffies et al., 2009).

Additionally, the sea-ice cover and its variability may affect the large-scale atmospheric circulation, also outside the high latitudes. For example, some studies suggest that the Arctic sea-ice loss has increased the frequency of atmospheric blocking events which then has changed the snowfall over America and Eurasia (Francis and Vavrus, 2012; Barnes, 2013). However, the

impacts of the Arctic warming on lower latitudes are masked by the large internal climatic variability and the detection of robust signals is very difficult due to relatively short time series of reliable observational data (Koenigk and Brodeau, 2016). These observational shortcomings can partly be overcome by analysing long climate model experiments which optimally should incorporate the most realistic sea-ice models to minimise the model uncertainty.

Recently, the version 3.6 of the Nucleus for European Modelling of the Ocean (NEMO) was released, along with its new sea-ice component, Louvain-la-Neuve Sea Ice Model (LIM) version 3.6 (Madec et al., 2015; Rousset et al., 2015). The new LIM3 code implements many sea-ice physics improvements compared to the previous LIM2 code, as has already been documented (Vancoppenolle et al., 2009b; Massonnet et al., 2011; Vancoppenolle et al., 2015). However, the effect of LIM3 on the ocean circulation and hydrography remain to be systematically investigated. Accordingly, our aim is to analyse NEMO-LIM2 and

NEMO-LIM3 simulations, including the most distinct ocean hydrography and circulation differences. As these differences may emerge over multi-decadal time scales due to slow oceanic processes, we carry out multi-decadal hindcast simulations with prescribed atmospheric forcing. This analysis assists us to comprehensively understand the oceanic response to the state-of-the-science sea-ice physics in multi-decadal ocean-ice hindcasts.

To support our task, a significant body of literature presenting ocean-ice model assessments provide us an important refer-
ence when carrying out our NEMO-LIM assessments. For example, papers of the CORE-II virtual special issue of the Ocean Modelling Journal, such as Danabasoglu et al. (2014, 2016); Downes et al. (2015); Farneti et al. (2015); Griffies et al. (2014); Wang et al. (2016a, b), and of the ORA-IP special issue of the Climate Dynamics Journal, such as Chevallier et al. (2015), are particularly relevant for this study. As the majority of CORE-II and ORA-IP ocean model configurations, our grid configuration (ORCA1) does not resolve ocean eddies. In this coarse-resolution ocean-ice model category, the eddy transport of

momentum and heat are parameterised, for instance. Our simulations also share the use of CORE bulk formulae with the CORE-II experiments (Large and Yeager, 2004).

In the polar context, which is the regional focus of our study, the most important CORE-II papers include Downes et al. (2015), where the Antarctic sea ice and Southern Ocean water masses are analysed, and Wang et al. (2016a, b) who investigated the Arctic sea ice and the Arctic Ocean freshwater. Recently, Chevallier et al. (2015) analysed the Arctic sea ice in a set of

ocean reanalyses to assess how the assimilation of observations affects the sea-ice characteristics. In addition to observational data, we use these ocean-ice model assessments as a benchmark when analysing our simulation performance.

The paper is divided into six sections. Section 2 describes the two versions of the ocean-ice models, NEMO-LIM2 and NEMO-LIM3, their initial and boundary conditions, model input data and observational reference data. In Section 3, we present sea-ice related results of the reference LIM3 hindcast simulation in comparison with observations and the reference

LIM2 hindcast. Section 4 presents results of the NEMO-LIM sensitivity simulations to test the robustness of LIM3 and LIM2 differences for surface freshwater adjustments. In section 4 we also assess how realistic sea ice a simplified LIM3 single-category ice thickness parameterisation reproduces. In section 5 differences of the ocean characteristics between NEMO-LIM2 and NEMO-LIM3 are discussed. Finally, the most important findings are highlighted in the conclusion section.

## 2 Models and Methods

All simulations presented here are based on the version 3.6_STABLE (revision 5918, released on 26 November 2015) of the NEMO-LIM ocean-ice modelling system (Madec et al., 2015), in the ORCA1 configuration. In NEMO, the OPA ocean component is coupled with the LIM sea-ice model. For almost a decade, LIM2 has been the reference NEMO sea-ice model (Fichefet and Morales Maqueda, 1997), but in June 2015, a new and more sophisticated version, LIM3.6, became available as the reference sea-ice model for NEMO3.6 (Rousset et al., 2015).

### 2.1 NEMO ocean component OPA

OPA is a finite difference, hydrostatic, primitive equation ocean general circulation model. Its vertical coordinate system is based on $z^*$ levels with partial cell thicknesses allowed at the sea floor. The vertical mixing of tracers and momentum uses the turbulent kinetic energy scheme (Gaspar et al., 1990; Blanke and Delcluse, 1993). A quadratic bottom friction boundary condition is applied together with an advective and diffusive bottom boundary layer for temperature and salinity tracers (Beckmann and Haidvogel, 1993). The model uses a non-linear variable volume scheme for the free surface, and an energy-enstrophy conserving scheme for momentum advection. A no-slip boundary condition is applied on the momentum equations with the horizontal Laplacian momentum diffusion. The tracer equations in OPA use the TVD advection scheme by Zalesak (1979) with the Laplacian diffusion along isoneutral surfaces.

The simulations are performed on an ORCA-like global tripolar grid with $1°$ nominal horizontal resolution and 75 vertical levels. Additional refinement of the meridional grid down to $1/3°$ is present near the Equator. A typical horizontal grid cell length is about 40–50 km in the Arctic Ocean and 40 km in the Antarctic region, while the vertical level thickness ranges from 1 m near the surface increasing to 200 m at the bottom.

### 2.2 NEMO sea-ice components LIM2 and LIM3

Our ocean-ice configurations only differ in their sea-ice component, all other experimental conditions being identical. We use the levitating sea-ice framework, following the convention of Campin et al. (2008): the growth and melt of ice impact the ocean mass and the salinity, but do not affect the pressure experienced by the ocean surface.

LIM2 (Fichefet and Morales Maqueda, 1997; Timmermann et al., 2005) is a sea-ice model in the line of the two-level model of Hibler (1979). It features: a single sea-ice category and open water represented using ice concentration; the Semtner (1976) 3-layer thermodynamics with a virtual reservoir of shortwave radiation heat which parameterizes brine inclusions; the revisited C-grid elastic-viscous-plastic rheology of Bouillon et al. (2013); the second-order moment-conserving advection scheme of Prather (1986), plus a few extra parameterisations. LIM2 implements the snow-ice formation by infiltration and freezing of seawater into snow when deep enough. The effect of sub-grid-scale snow and ice thickness distributions is implicitly parameterised by enhancing the conduction of heat through the ice and by melting the ice laterally to account for thin ice melting. The surface albedo depends on the state of the surface (frozen or melting), snow depth, and ice thickness following Shine and Henderson-Sellers (1985).

LIM3.6 (Vancoppenolle et al., 2009a; Rousset et al., 2015) is a sea-ice model in the line of the AIDJEX model, with multiple sea-ice categories (Coon et al., 1974; Thorndike et al., 1975). Multiple categories allow to resolve the intense growth and melt of thin ice, as well as the redistribution of thinner ice onto thicker ice due to ridging and rafting. LIM3 dynamics (advection and rheology) are the same as in LIM2. Thermodynamics are multi-layer and include an explicit description of the effect of brine on the storage and conduction of heat (Bitz and Lipscomb, 1999), and a parameterization of brine drainage (Vancoppenolle et al., 2009a) that affects ocean-ice salt exchanges. The default NEMO3.6 configuration uses five ice thickness categories and two vertical layers for thermodynamics. Alternatively, LIM3.6 (Rousset et al., 2015) can be run with a single sea-ice category, using two virtual ice thickness distribution parameterisations (enhanced conduction and thin ice melting).

## 2.3 Model input data

The NEMO model bathymetry is a combination of ETOPO1 Amante et al. (2009) in the open ocean, and GEBCO (IOC, 2003) in coastal regions. All the simulations were extended over the period 1958–2012 and forced by the DFS5.2 atmospheric data set, developed through the DRAKKAR consortium (Brodeau et al., 2010). This data set is based on satellite observations (monthly precipitations and daily radiative heat fluxes) and combined ERA-40 (before 1979) and ERA-Interim (from 1979 onward) meteorological reanalyses. DFS5.2 provides 3-hourly air temperature and humidity at the 2 m level, and wind velocity at the 10 m level (Uppala et al., 2005; Dee et al., 2011). Prescribed surface boundary conditions were calculated by using the CORE bulk formulae proposed by Large and Yeager (2004). As in Brodeau et al. (2010), simulations were forced with the monthly river run-off climatology based on Timmermann et al. (2009). The ocean and sea-ice models had a time step of 3600 s, which was also the interval when surface boundary conditions were updated.

As a standard practice in forced ocean-ice simulations, the mean sea level controls were used to prevent the unrealistic drift of the sea surface height due to freshwater boundary forcing distorted by errors in precipitation, evaporation and river runoff (Griffies et al., 2014). Specifically, this was done by setting values `nn_fwb=2` and `nn_ssr=.true.` in the NEMO namelist. The `nn_fwb` parameter is used to reset the freshwater budget, evaporation minus precipitation minus river runoff, and `nn_ssr` enables the restoring of sea-surface salinity (SSS). The SSS restoring rate is a negative feedback coefficient which is provided as a namelist parameter `rn_deds`. This parameter should be viewed as a flux correction on freshwater fluxes to reduce the uncertainties of the observed freshwater budget. Following the default ORCA1 NEMO3.6 configuration and discussions with NEMO users, the SSS restoring rate was set to -100 $mm/day$ towards the SSS of Polar Hydrographic Climatology version 3 (PHC3) created by Steele et al. (2001). Notably this is a lower value than the NEMO default of -166.67 $mm/day$.

## 2.4 Experiments

NEMO-LIM simulations started from the state of no motion in January 1958, with initial conditions for ocean temperature and salinity derived from PHC3 (Steele et al., 2001), and ended in December 2012. We completed two reference simulations, one using NEMO-LIM3 and another one using NEMO-LIM2, with recommended settings. In both LIM configurations, the initial sea-ice thickness was set to 3.0 m where the sea surface temperature was below 0 °C. The initial snow thickness was set to 0.3 m in LIM3 for both hemispheres, while in LIM2 it was 0.5 m in the Northern Hemisphere (NH) and 0.1 m in the Southern

Hemisphere (SH). For both simulations, the initial sea-ice concentration was set to 90%, except for LIM2 in the NH, where the initial sea-ice concentration was 95%. The ice strength parameter $P^*$ was set to $2\times10^4$ ($1\times10^4$) $\mathrm{Nm}^{-1}$ in LIM3 (LIM2), while the ocean-ice drag coefficient was $5.0\times10^{-3}$ and the atmosphere-ice drag coefficient follows Large and Yeager (2009) in both models.

Differences between LIM2 and LIM3 initial sea-ice and ice dynamics parameter values originate from the fact that they are recommended values according to the default NEMO3.6 configuration. Instead of setting the LIM2 initial values and ice dynamics identical to LIM3, for example, we took the point of view that we compare two different sea-ice models, each with its own specific and optimised tuning, and with no specific focus on ice dynamics. This is the point of view that was adopted by Massonnet et al. (2011) as well.

Even though the sea-ice initialisation differs slightly in terms of hemispheric snow thickness, it does not impact our results which focus on the last decade 45 years after the initialisation. The lower LIM2 ice strength $P^*$, however, has an impact and results in weaker ice that deforms easier producing a larger sea-ice volume than would have obtained with the $P^*$ value identical to LIM3. To quantify this effect, a LIM2 simulation with the LIM3 $P^*$ value produces a too small winter Arctic sea-ice volume, while its sea-ice extent remains almost unchanged (not shown here). Moreover, as found by Holland (2006),

multiple sub-grid-scale ice thickness categories, as in LIM3, reduces the effective ice strength compared to a single category model, as LIM2. Therefore, it is justified to use a higher $P^*$ in LIM3 to offset the reduces ice thickness–ice strength effect. According to these findings, LIM2 with the lower $P^*$ is more realistic and we decided to use it this study.

In addition to the two reference simulations, we carried out sensitivity simulations to determine how significant and systematic differences between LIM3 and LIM2 are. In these sensitivity simulations, processes related to ocean-ice interactions were

regulated and adjusted. In this way, we were able to isolate the impacts of individual processes and quantify their significance. First, we switched NEMO-LIM3 into its single-category mode which employs a virtual ice thickness distribution parameterisation, which make the model simpler and computationally cheaper than with multiple categories. Then, instead of using a prognostic salinity profile, we set the LIM3 sea-ice salinity to a constant value of 4 ppm, similarly to LIM2. As a result of the single-category and constant sea-ice salinity, LIM3 is physically to a greater extent closer to LIM2, is computationally fast, but

more realistic than LIM2, particularly in the Arctic, as we will show. Table 1 summarises these simulation characteristics.

The second set of sensitivity experiments were performed to examine the impact of ocean surface boundary conditions on ocean-ice properties, and therefore to see how robust our LIM3–LIM2 comparison results are. For this, we carried out NEMO-LIM2 and NEMO-LIM3 simulations where the mean sea level controls were switched off by setting `nn_fwb=0` and `nn_ssr=.false.` in the NEMO configuration namelist. After completing these NEMO-LIM2 and NEMO-LIM3 simula-

tions without freshwater adjustments, we calculated and compared their differences to the corresponding ones based on the reference simulations where the freshwater controls were kept on.

## 2.5   Reference data

When quantitatively assessing the modelled sea-ice and upper ocean realism, we included comparisons with satellite-based and reanalysis products of sea-ice concentration, thickness and velocity. Since 1979, space-borne passive microwave sensors have

produced a nearly continuous and consistent record of ice concentration which provides a good basis for model validation. For sea-ice concentration we used the NOAA/NSIDC Climate Data Record of Passive Microwave Sea Ice Concentration, Version 2 (Meier et al., 2013), which covers both polar regions at a 25 km grid cell size. The observed sea-ice extent data, which were based on satellite observed sea-ice concentrations, are the NSIDC Sea Ice Index (Fetterer et al., 2002). For sea-ice velocity analysis, the OSI-SAF product by Lavergne et al. (2010) was used. Sea-ice thickness and volume were compared with reanalyses from the Pan-Arctic Ice-Ocean Modeling and Assimilation System (PIOMAS) for the NH, and from the Global Ice-Ocean Modeling and Assimilation System (GIOMAS) for the SH (Schweiger et al., 2011). It is worth noting that these ice thickness data are model products, not entirely based on observations and have significant uncertainties.

For hydrographic comparisons, we decided to use two observational data sets. First, we selected aforementioned PHC3, which is a global climatology with a combination of NODC's 1998 world climatology, the EWG Arctic Ocean Atlas, and selected Canadian data provided by the Bedford Institute of Oceanography (Steele et al., 2001). PHC3 was updated with the Arctic Ocean temperature and salinity observations in 2005. Additionally, we used the most recent World Ocean Atlas 2013 (WOA13) (Boyer et al., 2013) averaged over the years 2005–2012. WOA13 lacks the Arctic observations included in PHC3, but has more recent observations elsewhere. Therefore, at least outside the Arctic Ocean, the WOA13 data temporally matches better with our NEMO-LIM comparison period of 2003–2012. As we will show in Section 5, WOA13 and PHC3 mainly differ in terms of Arctic SSS, while otherwise climatological differences are relatively small from the NEMO-LIM assessment perspective. Notably, Wang et al. (2016b) used PHC3 in their ocean-ice model Arctic intercomparison study, and using PHC3 here as well makes qualitative comparisons of our results with theirs straightforward. Finally, PHC3 was used to determine the initial hydrographic conditions of our model simulations, and comparisons between 2003–2012 LIM3 climatology and PHC3 show how much our simulations diverged from their initial state in 45 years.

Modelled mixed layer depths (MLD) were compared with the observational climatologies by de Boyer Montégut et al. (2004) and Holte et al. (2016). It should be noted that observational uncertainties in the Arctic Ocean and in polar regions south of 55°S are particularly large due to a limited number of measurements. Hence, we concentrated in the North Atlantic and the Southern Ocean outside the regions of high uncertainty in our MLD comparisons.

## 3 Sea-ice results

In this section, we analyse how well LIM reproduces large-scale climatological sea-ice properties (ice areal coverage, volume and drift). In order to evaluate the new sea-ice model, we compare LIM3 results to satellite observations, reanalysis data and as well as to the equivalent LIM2 simulation. All mean fields are computed over the last decade of integration, from 2003 to 2012. As the LIM3 sea-ice properties have already been analysed by (Vancoppenolle et al., 2009b; Massonnet et al., 2011; Vancoppenolle et al., 2015) and our results agree rather well with theirs, we rather shortly present our sea-ice findings and merely focus on novel findings from the sensitivity simulations and oceanographic analysis in the sections following this one.

## 3.1 Sea-ice concentration and extent

In the NH in September, the geographical distribution of LIM3 sea-ice concentration presents high values in the Canadian Arctic Archipelago with a realistic latitudinal decrease toward the Eurasian Arctic (Figure 1a). LIM3 tends to generally underestimate the ice concentration, by ∼20% in the central Arctic to ∼50% in the northern Kara Sea (Figure 1b), while the Laptev and southern Kara Seas are almost ice-free (Figure 1a). This negative summer sea-ice concentration bias is linked to an underestimation of sea-ice thickness in those areas both in winter and summer (not shown here). By contrast, too-large ice concentration is found in the Beaufort Sea (Figure 1b). Clearly the representation of ice concentration in the two models significantly differs in summer: LIM2 produces higher sea-ice concentration compared to LIM3 everywhere in the Arctic Ocean and their difference increases radially from the Canadian Arctic Archipelago toward the Eurasian Arctic (Figure 1e). LIM2 cannot reproduce the seasonal cycle of ice area in the Beaufort and East Siberian Seas toward the Bering Strait, where its sea ice, unrealistically, is rather uniform spatially with a too small open water fraction until a sharp transition to the ice edge (Figure 1c).

Mean seasonal cycles of the modelled sea-ice extents are shown in Figures 2a and c together with the NSIDC observations, all averaged over the years 2003–2012. In the NH, the LIM3 sea-ice extent closely follows the observed data and represents a clear improvement compared to LIM2, particularly in summer (Figure 2a). The respective LIM2 values are too high. LIM2 does not manage to melt enough ice and systematically overestimates the NH sea-ice extent. On the contrary, the LIM3 multi-category sea-ice thickness distribution allows for larger rates of melting due to its thin ice categories compared to the single-category LIM2, and enhances the seasonal cycle of sea-ice extent bringing it closer to observations.

Associated with the better mean seasonal cycle, the inter-annual time series of sea-ice extent is improved in LIM3 compared to LIM2 (Figure 2b). Both the maximum and minimum sea-ice extent are well reproduced by LIM3, as shown by the time series in Figure 2b that closely follow the NSIDC data in 1979–2012. Moreover, LIM3 realistically captures most of the summer minimum extents, including the 2007 record minimum. In contrast, LIM2 systematically overestimates yearly minimum, maximum and mean sea-ice extents during the whole period of integration. For example the 2007 minimum is overestimated by 50%. The two LIM models show comparable negative sea-ice extent trends in March, which are less negative than satellite observed trends. In September, the LIM3 trend is close to the observed one, while the LIM2 negative trend is too small. As concluded by Wang et al. (2016a), models which overestimate the Arctic sea-ice thickness, as does LIM2, have a too low September trend, while LIM3, which has a thinner ice, produces a realistic September trend.

In the SH, the LIM3 sea-ice edge is generally well located in the austral summer and the geographical distribution is correctly represented (Figure 1g, h). LIM3 sea ice is mostly confined to the western Weddell Sea, the southern Bellingshausen and Amundsen Seas and the southeast Ross Sea. Some differences with satellite observations are present. Notably, LIM3 underestimates the narrow fringe of sea ice around the East Antarctic coast and its sea ice also disappears excessively in the western Weddell Sea, where the model has a lower sea-ice concentration than observed, also indicating that its sea ice is too thin regionally. The LIM2 sea-ice concentration is everywhere larger than the LIM3 one and the observed one across most of the Southern Ocean, with the largest differences in the Ross Sea and the eastern Weddell Sea (Figure 1i, k).

Both LIM models have a seasonal cycle of sea-ice extent with too large amplitudes (Figure 2c). Periods of sea-ice growth are shorter, and sea-ice growth/melt rates are faster than observed. In LIM3, the monthly minimum sea-ice extent in February is less than the observed, while the maximum sea-ice extent in September is overestimated with a seasonal amplitude of $19.2 \times 10^6$ km$^2$ (observed $16.0 \times 10^6$ km$^2$). The LIM2 minimum extent appears to be in better agreement with the NSIDC data, but the ice growth is even faster than in LIM3, and therefore clearly unrealistic. As a result, the LIM2 seasonal cycle amplitude is $19.9 \times 10^6$ km$^2$.

The time series of annual mean sea-ice extent of LIM3 is rather well reproduced and closely follows observations (Figure 2d), but the sea-ice summer retreat is systematically too strong and summer extent too low. The LIM3 winter sea ice is on the average thicker than the LIM2 sea ice, while in summer their average thicknesses are close to each other (not shown here). On the other hand, the average LIM3 sea-ice concentration is systematically about 1–10% smaller than the LIM2 one, even in the central ice pack. As a result, the LIM3 sea-ice extent is smaller, particularly in summer.

The processes explaining the low LIM3 summer sea-ice extent compared to LIM2 are related to (1) the steeper decline of LIM3 mean sea-ice thickness and (2) to its systematically lower sea-ice concentration. Arguably the most important process is the positive ice-albedo feedback, which is governed by the fast melting of thin ice enabling an effective penetration of solar energy into the upper ocean. Negative sea-ice–related feedbacks are the ice thickness–ice strength relationship and the ice thickness–ice growth rate relationship, which is important during the growth period. These processes affect the ice evolution in both models. However, models with sub-grid-scale ice thickness distribution, as LIM3, have a less resistant ice pack to convergence resulting in thicker ice than a single-category modeli, as LIM2, under similar conditions (Holland, 2006). In LIM3, this feedback exposes more open water during the melt period. It is reasonable to assume that the primary reason for the LIM3 low summer sea-ice extent seems to be its lower sea-ice concentration to begin with, and consequently its large open water fraction, which reduces its grid cell mean albedo and enhances its ice-albedo feedback. The LIM3 sub-grid-scale ice thickness distribution further enhances the ice-albedo feedback, while simultaneously reducing the ice thickness–ice strength feedback.

In the SH in September, both LIM models present statistically significantly increasing sea-ice extent anomaly trends, estimated with the linear least-squares fit at the 5% significance level, consistent with observations. However, these modelled September trends are larger than the observed trend (not shown here). The increase of the Antarctic sea-ice extent has been explained by a range of mechanisms. Many studies attribute the increase of sea-ice extent to changes in the atmospheric dynamics, mainly by the increasing trend of the Southern Annular Mode, which in turn has strengthened westerly winds around the Antarctic continent and deepened the Amundsen Sea Low. Stronger westerlies effectively spread sea ice to north and a deeper Amundsen Sea Low increases the sea-ice production in the Ross Sea (see for example, Lefebvre and Goosse, 2008; Holland and Kwok, 2012; Massonnet et al., 2013; Turner et al., 2015). Another potential, simultaneously affecting mechanism increasing the sea-ice extent is the freshening of the Southern Ocean surface, which stabilises the surface layer, reduces the oceanic heat from below and therefore the associated ice melt (see for example Hellmer, 2004; Bintanja et al., 2013). Our model configurations do not implement the inter-annually increasing freshwater forcing, but despite that are able to reproduce the increase of winter Antarctic sea-ice extent. This implies that changes in windiness are likely to be a major mechanism driv-

ing the SH sea-ice extent increase. Notably, the LIM modelled trends are larger than the observed ones, which may indicate a too sensitive ice drift response to increasing windiness, a too fast moving model sea ice and a too far northern winter sea-ice edge, as also supported by earlier studies (Uotila et al., 2014; Lecomte et al., 2016).

## 3.2 Sea-ice volume

In the NH, the monthly mean LIM3 sea-ice volume, which is the domain integral of the sea-ice thickness multiplied by sea-ice area per grid cell, varies from the minimum of $8.8 \times 10^3$ km$^3$ in September to the maximum of $29.4 \times 10^3$ km$^3$ in April. Both values are approximately 20% larger than the PIOMAS output. LIM2 and LIM3 maxima agree, but their September minima do not, with the LIM2 ice volume minimum being almost 30% larger. The evolution of the annual mean sea-ice volume in the 1958–2012 period is comparable in both models and, as in the case of sea-ice extent, has large inter-annual variations (not

shown here). As with the sea-ice extent, NEMO-LIM simulations capture the large decrease of sea-ice volume during their last decade, 2003–2012, at a rate of -3.4 (-6.6)$\times 10^3$ km$^3$/decade in LIM3 (LIM2), while the PIOMAS rate is smaller, -2.0$\times 10^3$ km$^3$/decade.

In the SH, the LIM models' monthly mean sea-ice volume reaches its maximum in October and then decreases to 1,600 km$^3$ in February. The GIOMAS monthly mean sea-ice volume maximum occurs already in September from which it decreases to

15 2,800 km$^3$ in February. In general, the GIOMAS monthly mean sea-ice volume is higher than the LIM ones, with a distinctly different seasonal evolution. When comparing the LIM SH sea-ice volumes with GIOMAS, one should remember that the LIM3 SH sea-ice extent is smaller and closer to the observed than the one of LIM2. Hence, the LIM3 mean ice thickness, which is the ratio between sea-ice volume and extent, is larger and more realistic than the LIM2 mean ice thickness, because their sea-ice volumes are rather similar. LIM3 sea-ice volume growth rate is ~30% less than the GIOMAS one, but ~10%

higher than the LIM2 one. In both LIM models, the periods of ice growth are typically longer and periods of melt shorter than in GIOMAS.

As with the sea-ice extent in the SH, the annual mean LIM3 sea-ice volume has a statistically significant positive trend of 1.7$\times 10^3$ km$^3$/decade at the 5% significance level over the past decade 2003–2012. In contrast, the GIOMAS sea-ice volume trend for the same period is statistically significantly negative, -1.4$\times 10^3$ km$^3$/decade, while LIM2 has no statistically significant

trend. This indicates that, unlike in the NH, three models disagree in terms of the evolution of Antarctic sea-ice volume, at least for this particular time period.

There are important differences between PIOMAS and NEMO-LIM, explaining the systematic deviation of their sea-ice volume from each other. First, PIOMAS uses NCEP based atmospheric forcing compared to the DFS one used in the NEMO-LIM simulations. Second, PIOMAS assimilates sea-ice concentration and SST data, while the NEMO-LIM simulations do not.

Finally, PIOMAS ocean and sea-ice models and the computational grid are different from NEMO-LIM ORCA1 configurations along with numerous physical parameterisations implemented in the models.

In general, close similarities between the LIM2 and LIM3 sea-ice distributions in the SH emphasise the importance of the ocean model dictating the evolution of sea ice, while the level of sophistication of sea-ice model has a smaller importance. This is, at least partly, due to the divergent large-scale sea-ice motion where sea-ice deformation remains small (Uotila et

al., 2000). Therefore, different sea-ice deformation parameterisations in LIM2 and LIM3 have a lesser significance as in the relatively shallow Arctic Ocean. Another difference between LIM2 and LIM3 is related to the sea-ice thickness distribution parameterisation, which again has a smaller importance in the Southern Ocean than in the Arctic due to a smaller role of the ice-albedo feedback and the lack of surface melt ponds on the Antarctic sea ice compared to oceanic effects.

## 3.3 Sea-ice drift

The simulated March and September mean (2003–2012) sea-ice velocities are shown in Figure 3, together with the OSI SAF sea-ice drift product (Lavergne et al., 2010). Both LIM models realistically represent observed large-scale ice drift patterns, which are a direct response to the atmospheric circulation.

In the NH, the LIM3 mean drift pattern in March consists of an offshore motion over Siberian shelves (4–6 cm/s), the anti-cyclonic gyre in the Beaufort Sea (2–4 cm/s), the transpolar drift (4–6 cm/s) from the coast of Eastern Siberia to Fram Strait (Figure 3c). The ice drift through Fram Strait and in the East Greenland Current is particularly strong (16–20 cm/s), as well as the southward drift (14–16 cm/s) through Davis Strait. The Arctic sea-ice velocities in both LIM models are generally higher compared to satellite estimates, and the location of the centre of the Beaufort Gyre is displaced westward, toward the Chukchi Sea (Figure 3a, c and e). A similar positive sea-ice velocity bias was reported by (Chevallier et al., 2015) who analysed 14 ocean-ice reanalysis products. This discrepancy might be a result of a too high air-sea momentum flux driving the ice too fast and, on the shelf regions, due to the lack of a fast-ice parameterisation. On the other hand, the OSI SAF satellite derived sea-ice velocities may have high uncertainties over those regions of highly concentrated and slowly moving ice. Furthermore, the displacement of the Beaufort Gyre in LIM3 agrees with the positive sea-ice concentration bias of Figure 1b where relatively thick ice has been accumulated to (not shown here).

The two LIM models perform somewhat differently in terms of sea-ice speed, LIM2 sea ice being generally faster, in particular in the Beaufort Gyre (Figure 3e). The 10-year mean Arctic sea-ice velocity in March is 4.6 (4.8) cm/s for LIM3 (LIM2). Time series of area export through Fram Strait present similar variability in both LIM simulations. During the last simulated decade, the annual mean area fluxes through Fram Strait correspond to more than 10% of the winter ice covered area, being 0.86 (0.89) million km$^2$ in LIM3 (LIM2), both being comparable to estimates based on SAR data (Smedsrud et al., 2011).

In the SH, the LIM models feature similar and realistic looking distribution of the September ice drift (Figure 3). They show realistic patterns of the Weddell and Ross Gyres, the westward coastal and eastward offshore circumpolar currents. The observed OSI SAF drift generally compares well with the modelled ones in terms of their large-scale velocity field patters although the modelled speeds appear faster than observed, particularly along the ice edge. That suggests that LIM models simulate the Antarctic sea-ice drift reasonably well albeit somewhat too fast which seems to be a consistent ocean-ice model bias (Uotila et al., 2014).

As in the Arctic, the two LIM models have similar sea-ice velocity magnitude within the central ice pack, but larger differences appear close to the ice edge, where the LIM3 ice drift is ∼2 cm/s faster than LIM2, and in the coastal areas, where LIM2 speed is ∼2 cm/s faster than LIM3 (Figure 3f). LIM3 has a smaller ice extent and a lower ice concentration close to the ice

edge (not shown here). There the LIM3 ice motion is closer to the free drift and therefore faster that LIM2 ice motion. In the coastal areas, differences in ocean currents and ice deformation parameterisations are likely causes for the velocity differences between the LIM models. The horizontal, perpendicular-to-coast salinity gradient is stronger in LIM2 than in LIM3 in a way that LIM2 coastal surface waters are fresher, while off the coast LIM2 surface waters are saltier than in LIM3. This difference in the salinity gradient modifies the density gradient, coastal geostrophic currents and ice drift along the coast.

### 3.4 Sea-ice salinity

One important new feature in LIM3 is the prognostic sea-ice salinity compared to the constant 4 ppm sea-ice salinity in LIM2 (Vancoppenolle et al., 2009a). LIM3 explicitly includes the salt water entrapment and drainage in sea ice, where it also impacts on the ice thermodynamic variables such as the specific heat, conductivity and enthalpy. Furthermore, when snow-ice is formed by flooding and freezing of a relatively thick snow layer on top of ice, the LIM3 snow-ice becomes saline in contrast to the LIM2 fresh snow-ice. Vancoppenolle et al. (2009b) found that these improvements impacted on the LIM sea-ice volume, and that the LIM3 sea ice compared better with observations than the LIM2 sea ice.

It is reasonable to assume that to some extent the more realistic LIM3 sea ice might be due to the advanced salinity dependent halo-thermodynamics and a more realistic seasonal cycle of sea-ice salinity, and associated upper ocean freshwater fluxes. In winter, newly formed LIM3 sea ice preserves a higher salinity than in LIM2 (Figure 4). In contrast, in summer, the remaining LIM3 sea ice has a 2–4 ppm lower salinity than LIM2 in the Arctic (not shown here). However, during the Antarctic summer, the LIM3 sea-ice salinity stays relatively high, except at the ice edge (not shown here). This is due to the fact that even in summer air temperature remains at freezing over the coastal Antarctic seas. As in Vancoppenolle et al. (2009b), our simulations confirm that the LIM3 prognostic sea-ice salinity behaves realistically.

### 4 Sensitivity simulations

Based on rather descriptive analysis of differences between the LIM models, presented in the previous section, we have gained a relatively comprehensive understanding of how their global sea-ice distributions compare. In this section, we address what makes LIM3 sea ice different from LIM2 sea ice. Model grid and atmospheric forcing are identical, sea-ice differences can only arise from differences in sea-ice model physics parameterisations and these differences can be further amplified by ocean-ice feedback processes. To find out which parameterisations are of importance in producing LIM model differences, we performed and analysed some additional simulations.

### 4.1 NEMO-LIM3 single-category simulation

LIM3 differs from LIM2 in two important aspects: LIM3 has a multi-category sub-grid-scale sea-ice thickness distribution and multilayer halo-thermodynamics scheme with prognostic non-constant sea-ice salinity profile. We tested the effect of these parameterisations by carrying out a LIM3 simulation with a single-category sea-ice thickness distribution having a virtual ice thickness distribution and a constant 4 ppm sea-ice salinity. Importantly, by setting the LIM3 sea-ice salinity constant, along

with its two vertical ice layers and one snow layer, its thermodynamics scheme becomes similar to the LIM2 one. However, the initialisation procedure of LIM3 is different from the one used in LIM2, as explained in Section 2.3. We denote the LIM3 single-category simulation as LIM3SC.

In terms of NH sea-ice concentration and extent, LIM3SC is located between LIM3 and LIM2 (Figures 1d, 1f and 2b). In the SH, LIM3SC annual-mean sea-ice extent follows closely to the one of LIM2 (Figure 2b and 2d). However, the monthly sea-ice extent climatology of LIM3SC is distinctly closer to LIM3 and does not have the distorted shape of LIM2 monthly sea-ice extent climatology (Figure 2a and c). Furthermore, the summer minimum and winter maximum of LIM3SC sea-ice extents clearly differ from LIM2 ones. This result suggests that the use of the single-category and constant salinity parameterisations brings LIM3 sea ice closer to LIM2 output, as expected, but apparent differences remain.

The LIM3SC sea-ice volume relative to two other LIM simulations is more different in the SH than in the NH (not shown here). In the Southern Hemisphere, the LIM3SC sea-ice volume immediately diverts from LIM2 and LIM3, although its annual mean sea-ice extent remains rather close to LIM2 with a seasonal variability closer to the LIM3 one. It is possible that strong ocean-ice feedback processes in LIM3SC affect the melting and freezing rates during its first simulation year, and associated fluxes of salt and freshwater. This in turn modifies the upper ocean stratification and oceanic heat, which result in further differences in LIM3SC sea-ice volume that adjusts above the LIM2 level. The 20 cm thicker LIM3SC initial snow might have contributed to the differences in sea-ice thickness between LIM2 and LIM3SC by reducing the spring melt at the end of the first simulation year resulting in a relatively high sea-ice volume minimum in summer that persists through the simulation. After this, the high LIM3SC sea-ice volume seems to be in a balance with the upper ocean adjusted during the first years of the simulation.

In addition to sea-ice thermodynamics, the sea-ice salinity scheme modifies the ocean-ice salt and freshwater exchange, and upper ocean heat fluxes, which influence the evolution of sea ice. Compared to LIM2, the LIM3 multi-category sea-ice is saltier in winter due to its prognostic sea-ice salinity (Figure 4). This implies a smaller ocean-to-ice salt flux during freezing and a more stably stratified ocean surface layer, particularly in the Southern Ocean and in the Barents Sea where LIM3–LIM2 winter salinity differences seem particularly large (Figure 4). If the LIM3 prognostic salinity was of primary importance, we would expect a higher sea-ice volume in the LIM3 prognostic sea-ice salinity simulation than in the LIM3SC constant sea-ice salinity simulation due to smaller salt rejection rates and associated ocean convection in the Southern Ocean. As this is not the case, the importance of the sea-ice salinity scheme, in modifying the sea-ice evolution by affecting upper ocean heat fluxes, appears to be a secondary one compared to the effects of sea-ice salinity scheme on sea-ice thermodynamics and especially to the effects of sub-grid-scale ice thickness parameterisation.

## 4.2   Effects of freshwater adjustments

Following a common practise when carrying out forced ocean-ice simulations, we applied a fresh water budget adjustment and SSS restoring in our simulations. The freshwater budget, evaporation minus precipitation minus river runoff, was adjusted from the previous year's annual mean budget to zero at the beginning of each simulation year. Additionally, we added a SSS dependent flux correction term on freshwater fluxes. This flux correction term practically damps the model top-level salinity

towards the PHC3 top level salinity PHC3 everywhere, also under sea ice, in LIM2, LIM3 and LIM3SC simulations. These treatments prevent an unrealistic drift of the sea surface height due to errors in the prescribed freshwater budget components.

In addition to the common practise, we completed two otherwise identical integrations, one for LIM2 and one for LIM3, where we turned off the two freshwater adjustment mechanisms to see what kind of effect they have on our results. As expected, the ocean salinity drift became remarkable in the non-adjusted simulations, being strongest in the top layer, increasing its salinity by 0.4 psu in 54 years. This salinity change resulted in a global sea-level decrease of 8 m and also modified the ocean density structure. Related to this, a shallower mixed layer in the northern North Atlantic, a slightly weaker (1-2 Sv) Atlantic Meridional Overturning Circulation (AMOC), and a somewhat larger temperature drift were detected from the non-adjusted stimulations.

Perhaps interestingly and in contrast to the North Atlantic, the Southern Ocean mixed layers were deeper without freshwater adjustments (LIM3FW and LIM2FW). Importantly, for the scope of this study, the effects of freshwater adjustments on sea-ice evolution were minuscule. LIM models produced almost identical sea ice independent of whether the freshwater adjustments were turned on or off. Mutual oceanic differences between the LIM3 and LIM2 simulations and between the LIM3FW and LIM2FW simulations did not change drastically, as we will show. However, as 54-year simulation is rather short from the ocean circulation perspective, it is possible that in longer simulations the differences between the simulations in terms of oceanic circulation increase to the point that they start to modify the sea-ice characteristics remarkably.

## 5 Ocean hydrography and circulation

### 5.1 Arctic surface salinity

We now move on to explore differences in ocean properties between the two LIM versions. Figure 5 shows LIM3 Arctic SSS and sea-surface temperature (SST) averaged over the last decade of the simulation, 2003–2012, and LIM3 and LIM2 departures from PHC3, WOA13 and LIM2. LIM models SSS is closer to that of PHC3 than WOA13 due to their surface salinity restoring towards PHC3. LIM models' differences from PHC3 and WOA13 are much larger than their mutual differences, highlighting the fact that the LIM version has a secondary impact on the Northern Hemispheric ocean properties.

LIM3 surface salinity distribution realistically reflects the fact that the Arctic Ocean has a low salinity surface layer in contrast to the much saltier surface layer of the North Atlantic (Figure 5a). Compared to PHC3 and WOA13, both NEMO-LIM versions are too fresh in the North Atlantic, Labrador Sea and Nordic Seas, although the LIM3 Labrador Sea surface saltier than that of LIM2, in particular close to the Greenland coast (Figure 5b–f). In some parts of the Eurasian Basin, LIM3 is saltier than PHC3 which is partly associated with its negative sea-ice concentration bias and the lack of fresh melt water (Figure 5b). Compared to WOA13, LIM3 SSS is much higher due to the SSS restoring toward PHC3, which indicates disagreements in terms of Arctic SSS due to the lack of observations and that the PHC3 SSS is higher than WOA13 SSS. As PHC3 was carefully constructed for the Arctic, it is plausible that its SSS climatology is closer to the truth. LIM ocean salinity biases mainly arise from the NEMO ocean model configuration, and the applied boundary conditions, such as the atmospheric forcing, river runoff and freshwater adjustments.

LIM3 has a fresher surface than LIM2 in many areas on the Siberian shelf, Barents Sea and Greenland Sea, associated with the smaller ice-ocean salt flux, thicker ice in winter and larger melt rates during spring (Figure 5d). By contrast, in LIM2, fresher ice and reduced spring melt result in an increased ice-to-ocean salt flux and therefore higher SSS in those regions. However, along the East Greenland coast, thicker LIM2 sea ice is associated with higher melt rates and result in a fresher surface also in the Labrador Sea, where the ice and freshwater drifts to. These differences in surface salinity, associated with sea-ice differences, have potential implications for the strength of AMOC. Hence, although mutual hydrographic differences between the freshwater adjusted LIM simulations are small compared to their observational biases, they may potentially have an impact on the convective processes in the North Atlantic.

The experiments without freshwater adjustments, LIM2FW and LIM3FW, display larger SSS differences with the expanded or changed regions of statistically significant difference (not shown here). For example, the region north of Greenland in the Arctic Ocean, which is significantly saltier in LIM3 than in LIM2 (Figure 5d), is not significantly saltier in LIM3FW than in LIM2FW. However, LIM3FW–LIM2FW SSS differences remain clearly smaller than the corresponding SSS differences with the observational climatologies. In general the geographical patterns of SSS differences remain rather similar, mainly the magnitudes of SSS differences change. In any case, this indicates that the freshwater flux corrections reduce the salinity differences originating from two different sea-ice models.

## 5.2  Arctic surface temperature

As with SSS, SST differences between the LIM models in the Arctic are small compared to their differences from PHC3 and WOA13 (Figure 5h–l). The LIM models have a distinct cold bias in the North Atlantic and in the Greenland Sea. These cold biases are related to the common atmospheric ERA-Interim based forcing which is known to have a cold anomaly over the Fram Strait–Svalbard region (Notz et al., 2013). In the Norwegian and Barents Seas, LIM models' surfaces are warmer than PHC3, while their differences to WOA13 are relatively small. The WOA13 climatology represents years 2005–2012 and better matches the NEMO-LIM analysis period of 2003–2012 than PHC3, which contains observations before 2005. As PHC3 SST is colder than WOA13 and LIM SSTs in the Norwegian and Barents Seas, it is evident that these differences are related to the recent warming of these regions.

Related to the smaller LIM3 sea-ice extent, LIM3 SST is warmer than LIM2 SST across most of the Arctic Ocean, along the East Greenland coast, in Baffin Bay and in the Labrador Sea. In contrast, SST in the Norwegian Sea and Barents Sea is lower in LIM3 than LIM2, associated with a lower salinity. In these regions, saltier LIM2 surface waters release less heat to the atmosphere before reaching the critical density and sinking down, which explains the warmer LIM2 SST. Additionally, LIM3 and LIM2 mixed layer depths show remarkable differences across the Nordic Seas, as we will soon show. These SSS, SST and MLD differences signify the varying locations of effective upper ocean convection in the LIM simulations.

## 5.3  Southern Ocean surface salinity

In the SH, LIM3/LIM2 and PHC3/WOA13 SSS differences are smaller than in the NH (Figure 6b, c, e, f). In the regions covered by sea ice, LIM models' ocean surfaces are fresher than PHC3, except in some areas along the East Antarctic coast.

These coastal differences are smaller between the LIM models and WOA13 than between the LIM models and PHC3 (Figure 6b, c, e, f), which could be related to a larger number of better quality coastal Antarctic observations included in WOA13 and the fact that the simulation analysis period temporally better matches with WOA13 than PHC3. LIM3 SSS differences with LIM2 have smaller magnitudes than the LIM models' differences with the observational climatologies (Figure 6d). Now, LIM3 Antarctic sea ice is less extensive, but thicker than LIM2 sea ice, on the average. Hence, off the Antarctic coast where the ice melts, more fresh water is released per area in LIM3 than in LIM2 resulting in a lower LIM3 SSS. Close to the Antarctic coast the LIM3 ocean surface is saltier than the LIM2 ocean surface. This is likely to be due to the greater winter ice formation rates in LIM3, associated larger salt flux from ice to ocean.

Processes related to the LIM3/LIM2 and PHC3/WOA13 SSS discrepancies in the Southern Ocean are likely to be associated with the other freshwater sources rather than the sea ice related fresh water exchange. Again, this is because the LIM and PHC3/WOA13 differences are of larger magnitudes than the differences between two LIM models. Most important external freshwater sources in the Southern Ocean are precipitation and melt water fluxes from the Antarctic continental ice sheet, and both of these sources are known to have large uncertainties. Given these observational freshwater and SSS uncertainties, we can expect significant differences between NEMO-LIM and PHC3/WOA13 ocean surface characteristics. However, as the NEMO-LIM simulations applied the SSS restoring, LIM and PHC3 sea-surface salinities did not evolve very far apart compared to the simulations where the freshwater was not adjusted (not shown here). As for the Arctic, the LIM experiments without freshwater adjustments display larger mutual differences although their geographical patterns do not essentially change (not shown here).

## 5.4 Southern Ocean surface temperature

The surfaces of the LIM simulations are colder than PHC3 and WOA13 around the East Antarctic and in the Ross Sea (Figure 6h, i). As these differences are associated with fresher surface and lower than observed ice concentration, it is likely that the more stable LIM surface stratification decreases the upward oceanic heat flux and increases the surface heat loss to the atmosphere due to larger open water areas. Consistent with this explanation, the somewhat higher LIM2 sea-ice concentration and SSS seem to result in a higher SST than the one of LIM3 around the East Antarctic (Figure 6j).

## 5.5 Arctic intermediate water (AIW)

In the Arctic Ocean, approximately at 250 m depth, lies the relatively warm AIW layer that originates from the Atlantic Ocean (Figures 7 and 8). AIW is below the halocline and therefore saltier than waters above it (Figure 8b). The NEMO-LIM models simulate too fresh and cold waters at 250 m in the Arctic Ocean (Figure 7b-c, h-i). During the first decade of NEMO-LIM simulations, their AIW layers cool and possibly due to too vigorous mixing lose heat to the water masses above resulting in weaker and broader thermoclines than PHC3 and WOA13 (Figure 8a).

LIM3 AIW remains warmer than LIM2 (Figure 8a), which indicates a somewhat larger Atlantic warm water inflow into the Arctic Ocean (Figure 7j). In the Nordic Seas and Barents Sea, LIM3 and LIM2 are warmer than PHC3 and WOA13 at 250 m (Figure 7h, i). This indicates that not enough oceanic heat enters the Arctic Ocean in the LIM simulations possibly due to their coarse model grid that does not adequately resolve the basin topography and eddy heat transport. The lack of warm Atlantic

water inflow to the Arctic and associated biases are consistent with the ones founded in the multi-model study by Wang et al. (2016b).

## 5.6 Temperature and salinity differences outside the Polar Regions

Outside the polar regions, small temperature and salinity differences emerge as the LIM simulations proceed. For example, LIM3 has a saltier Atlantic Ocean than LIM2 at the layer from the surface to 1000 m depth (not shown here). LIM salinity differences vary in time with maximum values up to 0.05 psu, while their Atlantic basin averaged salinities are approximately 35.35 psu. On the basin scale, salinity differences between the two LIM simulations above 1000 m depth become notable rather soon, during the first decade of simulations. In other basins, salinity and temperature differences above 1000 m depth and outside the polar regions remain much smaller. However, in the Atlantic Ocean in the layer deeper than 1000 m, LIM3 starts to evolve into a fresher and colder state than LIM2 from the late 1970s onward. There basin-wide mean salinity differences amount up to 0.001 psu by the end of simulations. At the same time, in the layer deeper than 1000 m of the Pacific Ocean, LIM3 becomes on the average saltier and colder than LIM2. Changes in these deep water characteristics originate from the surface perturbations, which are slowly transported deeper by the meridional overturning circulation and deep water formation. The key regions where atmosphere-ocean conditions permit the deep water formation and consequently drive the meridional overturning circulation are located in the northern North Atlantic and coastal Antarctica. These are also the regions where the sea-ice cover between the two LIM simulations vary and thus modify the atmosphere-ocean energy exchange, which then affects the deep water formation and the World Ocean meridional overturning circulation.

## 5.7 Mixed layer depth (MLD)

Oceanic convection, vertical heat transport and deep water formation are intimately related to the MLD. We keep in mind that the observational MLD uncertainties are particularly large in ice covered oceans, because of sparse observations, and therefore limit our comparisons to the North Atlantic and the Southern Ocean. In Figure 9, the mean winter MLD is presented for LIM3 along with its difference from the observed climatologes of de Boyer Montégut et al. (2004) and Holte et al. (2016), LIM2 differences from the observed climatologies and for the LIM3–LIM2 difference. Clearly, the regions of deep MLD are located off the sea-ice edge (Figure 9a, g).

In the Norwegian Sea and the Barents Sea the observational climatologies agree generally rather well, but in the Labrador Sea where Holte et al. (2016) show deeper MLDs than de Boyer Montégut et al. (2004). In the North Atlantic, the LIM3 and LIM2 MLDs are larger than the observational MLD estimates indicating stronger oceanic convection (Figure 9b, c, e, f). This is at least partly due to the cold, non-responsive prescribed winter atmosphere acting as an infinitive heat sink to the relatively warm ocean surface layer and associated with relatively cold LIM3 and LIM2 SSTs (section 5.2).

In the Southern Ocean, the LIM3 and LIM2 mixed layers are again much deeper than the observed mixed layers outside the regions covered by ice, particularly in the Pacific sector (Figure 9h, i, k and l). This pattern, to some extent, resembles the MLD difference pattern in the North Atlantic. In the Southern Ocean, as in the North Atlantic, the very deep NEMO-LIM MLD is a

likely response to the prescribed atmospheric forcing and possibly to erroneous precipitation and freshwater fluxes originating from the Antarctic ice sheet.

Although the LIM3 MLD generally appears to be relatively close to the LIM2 one in the Arctic, it is statistically significantly deeper in some Arctic Ocean locations, such as in the Canadian Basin, where the LIM3 ocean surface is somewhat warmer and fresher (Figures 9d, 5d and 5j). In the Southern Ocean, LIM2–LIM3 MLD differences are quite small, with the statistically significantly regions visible in the marginal ice zone off the coast of east Antarctica and in along the Antarctic coast (Figure 9j). As these coastal regions are the Antarctic Bottom Water formation regions, MLD differences indicate differences in the locations and rates of the deep water formation between the two LIM simulations. This variability in the deep water formation changes the deep water properties, which is manifested as slowly emerging differences in abyssal temperature and salinity, decades after the beginning of simulations, as discussed earlier in section 5.6.

## 5.8 Atlantic Meridional Overturning Circulation (AMOC)

An important characteristic of a global ocean model is the strength and extent of its AMOC. The observational based estimates of its average strength vary between 16.9 Sv at 26.5°N (Smeed et al., 2016) to 18.5 Sv at 24°N (Ganachaud, 2003; Lumpkin and Speer, 2007) and 16.5 Sv at 48°N (Ganachaud, 2003). In terms of modelled AMOC, Danabasoglu et al. (2014) assessed the mean AMOC of eighteen ocean-ice models forced by prescribed atmospheric forcing from 1948–2007 and ran five repetitive forcing cycles, restarted from the state at the end of the previous cycle. Many of these fifteen ocean models were also used as the ocean-ice components of CMIP5 climate models. Here, the results of Danabasoglu et al. (2014) provide a useful indicative benchmark for our NEMO-LIM simulations, although one needs to keep in mind differences between CORE-II and our experiment setup, which are likely to affect AMOC. In Figure 10, the time evolution of NEMO-LIM AMOCs look rather similar to some ocean-ice models assessed by Danabasoglu et al. (2014) during their first simulation cycle (compare Danabasoglu et al. (2014) Figure 1, middle panel and our Figure 10a). As the NEMO-LIM models studied here, these models initially have an AMOC of 16-17 Sv which gradually, during the first three decades, decreases down to 8-12 Sv. These models, labelled as NOCS, CERFACS and CMCC in Danabasoglu et al. (2014), are based on earlier versions of NEMO than NEMO3.6, but importantly share the identical ORCA1 horizontal grid with us. The NOCS model has 75 vertical levels, as our NEMO-LIM configurations, while CERFACS and CMCC have a smaller number of vertical levels.

In addition to AMOC temporal evolution, our mean AMOC transport patterns in depth-latitude space well resemble the NEMO ORCA1 ones of Danabasoglu et al. (2014). This can be seen by comparing our Figure 10c with their Figure 3. In particular, NOCS and CMCC mean patterns resemble our NEMO-LIM3 pattern with high northward transport regions at 1000 m, a surface maximum at 10-20°N, and a rather strong southward transport approximately at 4000-5000 m (not shown). These qualitative inspections with Danabasoglu et al. (2014) indicate that our NEMO- LIM simulations produce a comparable AMOC to earlier NEMO configurations with comparable horizontal and vertical resolutions.

Deviations between the LIM2 and LIM3 simulations in terms of their AMOC are minor. There are, however subtle, statistically non-significant, differences, as seen from Figure 10a, where the LIM2 annual maximum AMOC within the 50-53°N band is up to 0.4 Sv stronger than the LIM3 AMOC within the same latitude band. The stronger LIM2 AMOC is likely to be

driven by stronger deep convection at the varying locations across the Nordic Seas (Figure 9d). Differences in MLD are related to differences in ocean surface stratification caused by deviations in sea-ice characteristics between the two LIM simulations.

As expected, the AMOC differences between LIM3 and LIM2 become more apparent when comparing simulations without freshwater adjustments, LIM3FW and LIM2FW. Both LIM3FW and LIM2FW have a statistically significantly lower AMOC at the 5% level than the ones with freshwater adjustments, LIM3 and LIM2 (Figure 10a). As the LIM3 AMOC is on the average smaller than the LIM2 AMOC, also the average LIM3FW AMOC is smaller (by 0.7 Sv n 2003–2012) than the average LIM2FW AMOC. However, the average LIM3–LIM2 AMOC difference is not statistically significant, while the average LIM3FW–LIM2FW AMOC difference is (at the 5% level). Accordingly, it is reasonable to assume that the freshwater adjustments, mainly the SSS-restoring, keeps the LIM3 and LIM2 AMOCs closer to each other.

## 5.9 Other oceanic transports

In addition to AMOC, we calculated time series of volume, heat and salinity transports through a number of oceanic transects: the Australia–Antarctica transect, the Bering Strait, the Denmark Strait, the Drake Passage, the Florida Strait, the Gibraltar Strait, and the Greenland–Norway transect at 60°N. LIM simulations show slightly varying volume transports in these major transects, such as the Drake Passage which we decided to show here (Figure 10b). There, LIM2 volume transports became approximately 5 Sv larger than in LIM3. These are relatively small deviations, given the fact that total volume transports in the Drake Passage are around 160 Sv. However, it is possible that these deviations further increase during long, multi-centennial simulations, as demonstrated by Danabasoglu et al. (2014) with respect of AMOC.

## 6  Conclusions

A set of hindcast simulations (1958–2012) was performed with the newest NEMO3.6 model using the global ORCA1 grid forced by the DFS5.2 atmospheric data. The primary objective was to diagnose the sensitivity of the NEMO-LIM ocean-ice system to the representation of physics in the sea-ice model. Results of such analysis have not been published for the newest NEMO in the nominal 1° latitudinal resolution, which is used as the ocean-ice component in many climate models participating in the CMIP6 project. We focussed on two simulations that differ only in their sea-ice component: the widely-used LIM2 and its successor, LIM version 3.6. The main differences between the two sea-ice models lie in their parameterisation of sub-grid-scale sea-ice thickness distribution, ice deformation, thermodynamic processes, and sea-ice salinity.

To assess the performance of two LIM versions, we compared their climatological sea-ice distributions mutually and with observational estimates. In terms of global sea ice, LIM3 compares clearly better with available observations, while LIM2 deviates more, producing too much ice in the Arctic, for example. The better representation of the ice-albedo feedback makes LIM3 more capable in simulating the September minimum of extent than LIM2, including the 2007 extremely low Arctic value. These sea-ice findings are consisted with the ones of Vancoppenolle et al. (2009b); Massonnet et al. (2011); Vancoppenolle et al. (2015); Rousset et al. (2015).

We mostly restricted our analysis to the last decade of the 54-year simulations. By analysing a ten-year period means that the effect of multi-decadal variability is not taken into account. However, earlier and longer analysis periods would have been more impacted by the model spin up from its initial state in 1958. Looking at the multi-decadal sea-ice extent and ocean transport time series (Figures 2 and 10), the LIM2 and LIM3 simulations stay systematically apart. Accordingly, it is sensible to assume that the respective LIM3–LIM2 differences are not very sensitive to the multi-decadal variability, at least during the last few decades of the simulations. Furthermore, it is reasonable to assume that the upper ocean LIM3–LIM2 differences behave like the sea-ice and the oceanic transport ones.

It is worth noting that no specific NEMO-LIM tuning was done for our experiments. It is likely that after some adjustments, such as controlled changes in the sea-ice albedo or ice strength, the NEMO-LIM3 and NEMO-LIM2 sea-ice performance will to some extent increase (Uotila et al., 2012). It is also worth noting that NEMO-LIM3 has been developed further and, for example, a new sea-ice albedo scheme was implemented in April 2016. This new scheme provides better transitions between the different ice types, slightly modifies the surface albedo compared to the old scheme and affects the model behaviour to a limited extent only. Hence, the results of this study remain valid even after the implementation of the new albedo scheme.

Our model evaluation focussed on the upper ocean properties and to some extent oceanic transports across major transects of the World Ocean, such as the Drake Passage, along with its meridional overturning circulation. This has not been systematically done before for NEMO3.6. In general, ocean hydrographic differences, such as temperature and salinity, between the two LIM versions are confined to the upper ocean and near the sea-ice zone. In terms of large-scale ocean circulation, differences between the two LIM versions remained small, but kept increasing over the decades, also in the extra-polar regions.

As a further sensitivity experiment, we repeated the NEMO-LIM3 hindcast simulation after setting its sea-ice distribution to the single-category mode. At large and as expected, this single-category configuration resulted in a shift of LIM3 sea-ice distribution towards the LIM2 one, but encouragingly the LIM3 single-category sea ice remained clearly more realistic than the LIM2 one. This result indicates that one option for modellers who are considering in upgrading from LIM2 to LIM3, is to start using the single-category LIM3 as an intermediate step. Based on these findings, we conclude that NEMO3.6 is ready as a stand-alone ocean-ice model and as a component of coupled atmosphere-ocean models.

# 7   Code and data availability

The NEMO version 3.6 version incorporates LIM2 and LIM3.6 sea-ice models, and can be downloaded from the NEMO web site (http://www.nemo-ocean.eu/) at this address: http://forge.ipsl.jussieu.fr/nemo/svn/branches/2015/nemo_v3_6_STABLE. The model input data can be obtained following the references described in section 2.5. The output of model simulations and the computer scripts used to produce the results presented in the paper, including the figures, are available from the corresponding author upon request.

*Acknowledgements.* We acknowledge the creators of low resolution sea-ice drift product of the EUMETSAT Ocean and Sea Ice Satellite Application Facility (OSI SAF, www.osi-saf.org). We thank Dr Laurent Brodeau for providing us the very useful Barakuda software package to diagnose NEMO simulations. The work of Lensu and Uotila was supported by the Academy of Finland (contracts 264358 and 283034).

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

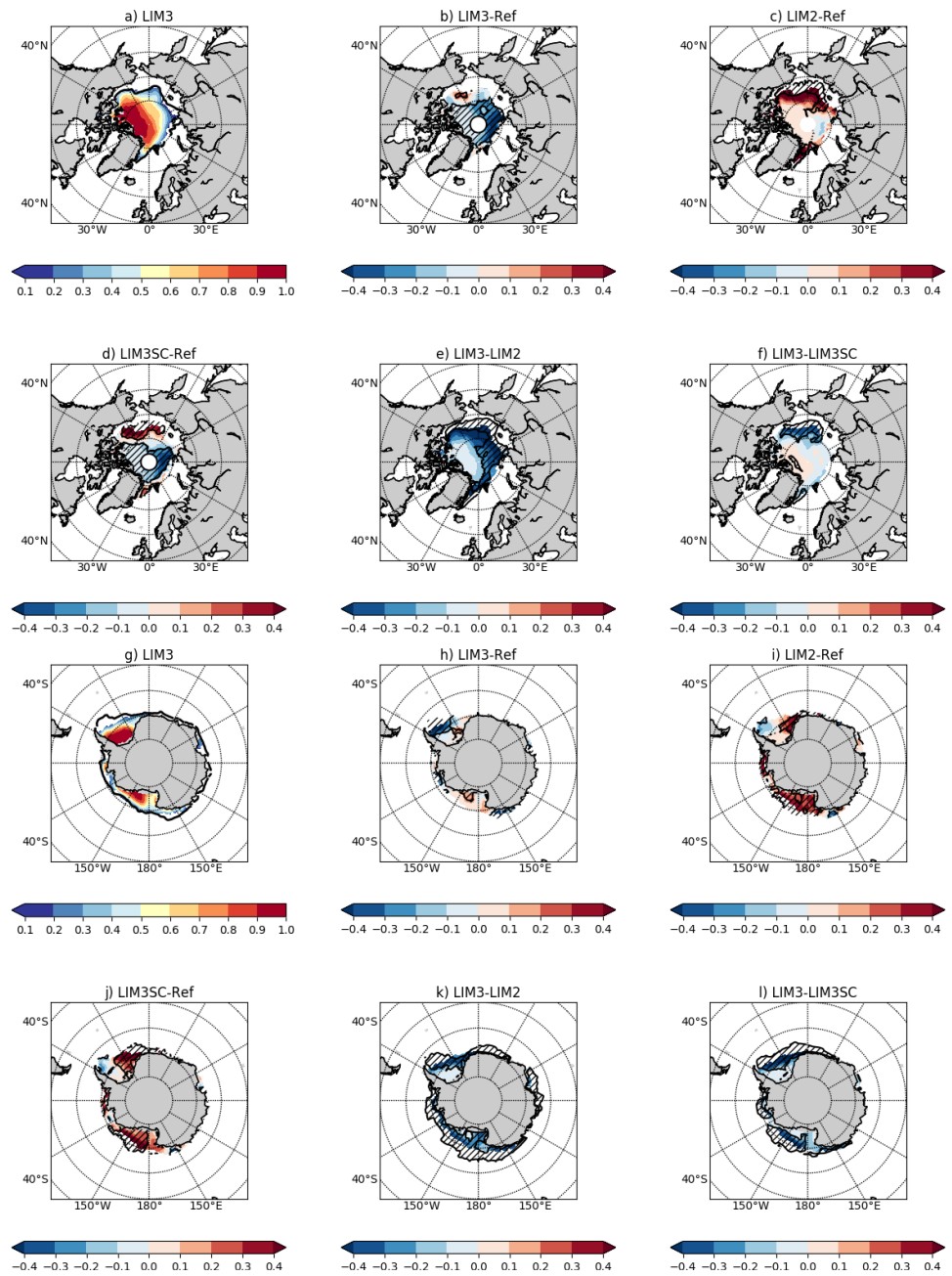

**Figure 1.** Geographical distribution of Arctic sea-ice concentration averaged for September (a–f) and Antarctic sea-ice concentration averaged for March 2003–2012 (g–l). (a, g) show values simulated by LIM3, while (b, h) show LIM3 difference with Meier et al. (2013) passive microwave observations, (c, i) the corresponding LIM2 difference, (d, j) the LIM3SC difference, while (e, k) show the difference between LIM3 and LIM2 and (f, l) the difference between LIM3 and LIM3SC. Sea-ice concentration differences are computed only where both values are present. Only areas where the sea-ice concentration is greater than 15% are plotted. In (a, e), thick black lines show the observed Meier et al. (2013) sea-ice edge as the 15% sea-ice concentration isopleth. Hatching indicates regions with statistically significant differences at the 5% level, based on unequal variances t-test.

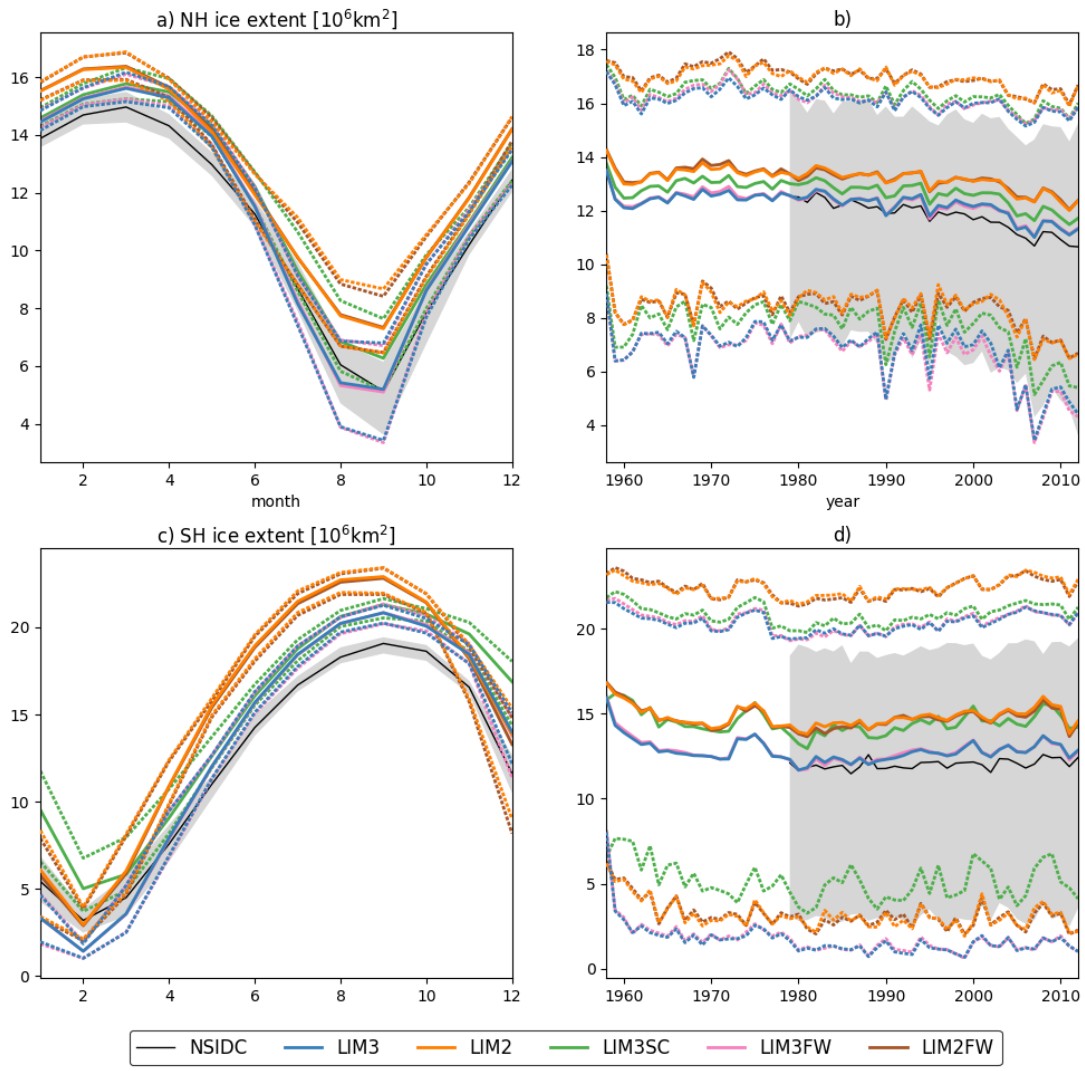

**Figure 2.** Simulated (coloured lines) and observed (black lines and grey shadings; NSIDC Fetterer et al., 2002) mean seasonal cycle (a, c) of monthly mean sea-ice extent over the period 2003–2012, for the (a) northern (NH) and (c) southern (SH) hemispheres. The sea-ice extent is calculated as the area with sea-ice concentration 15% or more. Dashed lines and grey shadings denote the minimum and maximum annual monthly extents during the same period. In the rightmost panels (b, d), annual maximum, mean and minimum time series of simulated and observed sea-ice extents in (b) the NH and (d) SH over the period of 1958–2012 are presented. LIM3 (blue lines) denote the reference LIM3 simulation, LIM2 (orange lines) denote the reference LIM2 simulation, LIM3SC (green lines) denote the LIM3 single-category sea-ice simulation, LIM3FW (magenta lines) denote the LIM3 simulation without the freshwater flux adjustments, and LIM2FW (brown lines) denote the corresponding LIM2 simulation.

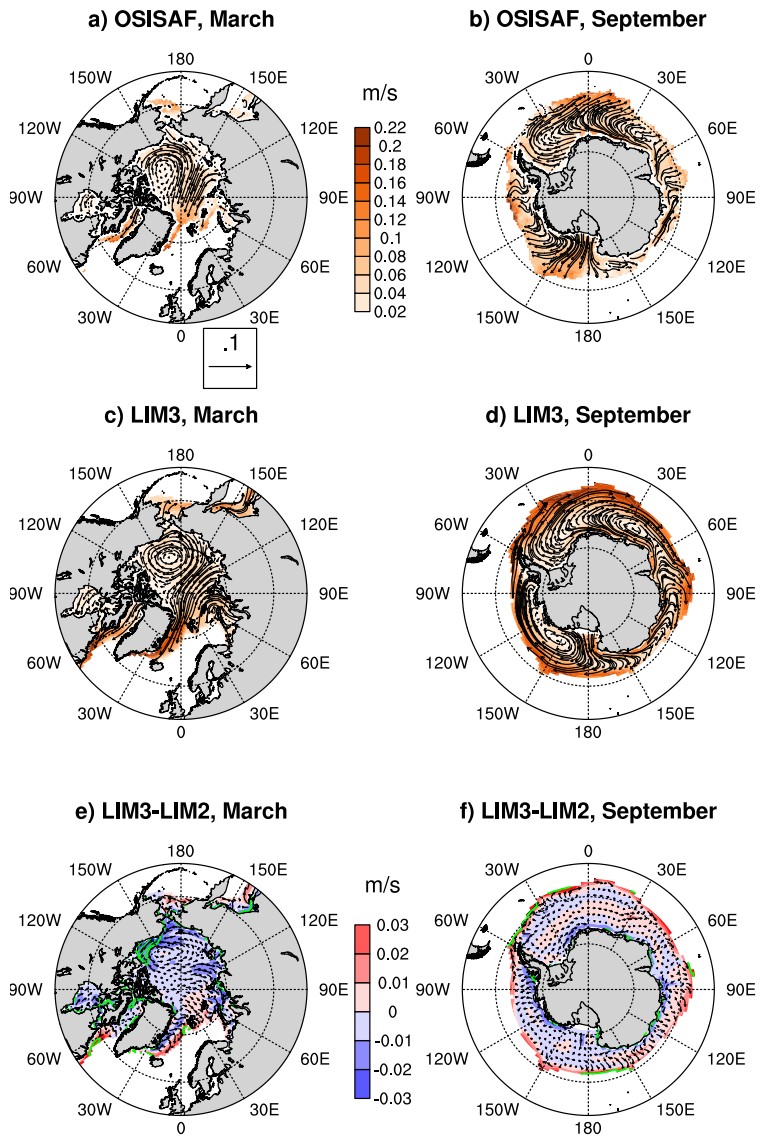

**Figure 3.** (a) Observed satellite-based average Arctic sea-ice velocity in March (Lavergne et al., 2010) as arrows and the corresponding vector magnitude (speed, m/s) as filled coloured contours based on years 2009–2015. (b) as (a), but for the Antarctic in September and based on years 2013–2015. (c) is similar to (a), and (d) is similar to (b), but for LIM3 ice velocity and speed based on years 2003–2012. In (e), mean differences between LIM3 and LIM2 ice velocity and speed are shown in the NH in March, while (f) displays the corresponding differences in the SH in September. In (e) and (f) green hatched regions show areas with statistically significant differences between LIM3 and LIM2 sea-ice speed at the 5% level based on unequal variances t-test.

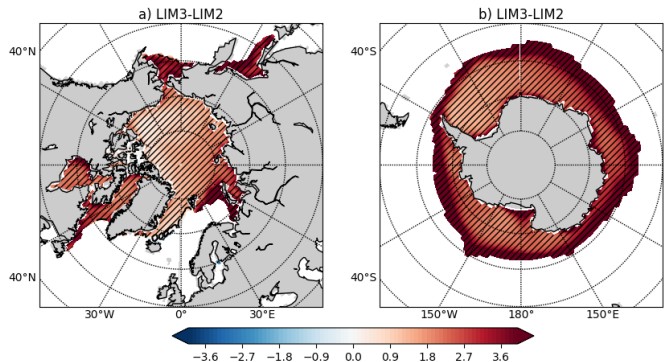

**Figure 4.** The sea-ice salinity difference (in ppm) between LIM3 and LIM2 (a) in the Arctic in March and (b) in the Antarctic in September. Note that the LIM2 sea-ice salinity is constant 4 ppm. Hatching indicates regions with statistically significant differences between LIM3 and LIM2 at the 5% level based on unequal variances t-test.

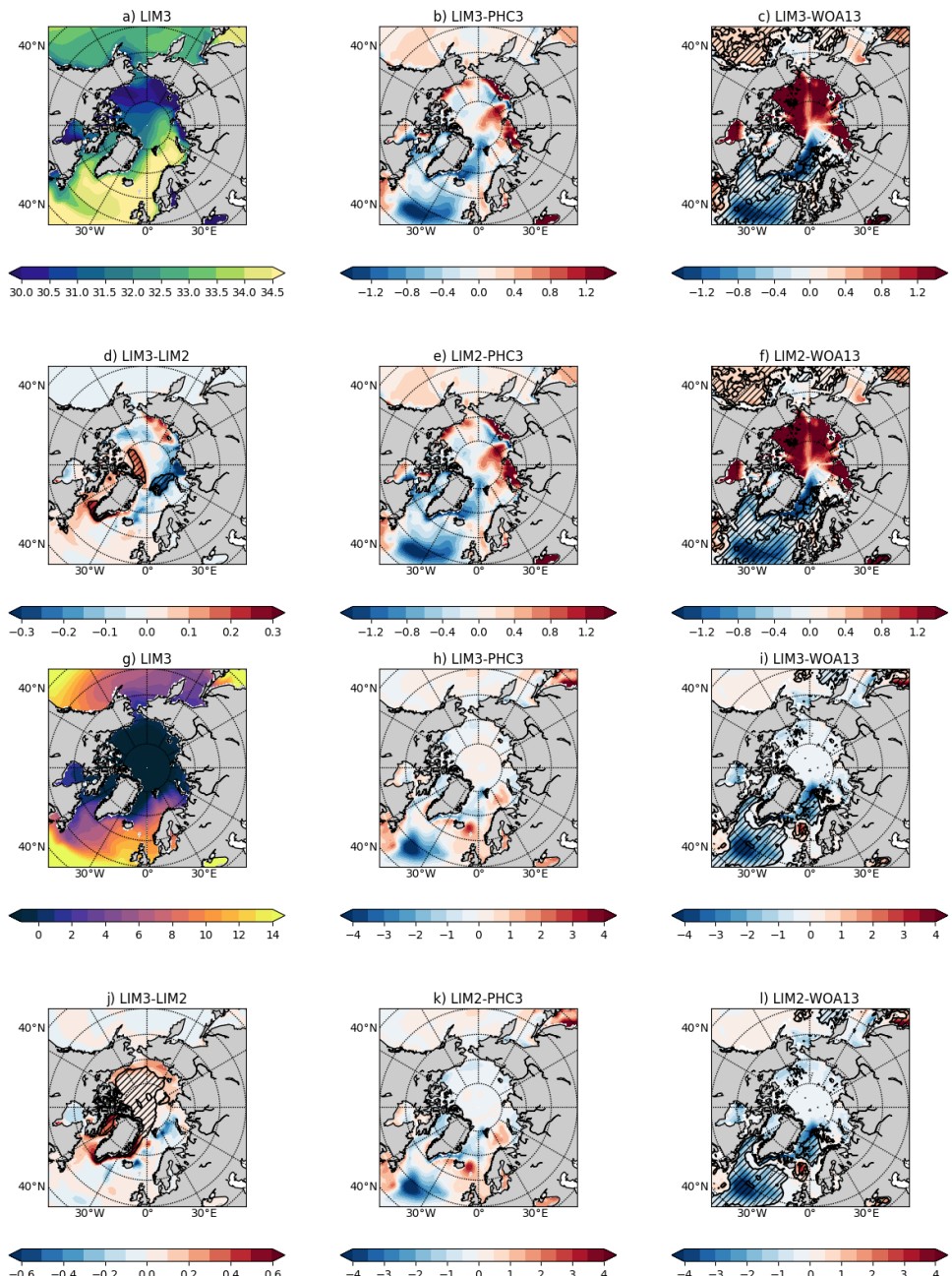

**Figure 5.** (a–f) Arctic sea-surface salinity in psu (SSS) and (g–j) sea-surface temperature in °C (SST) averaged over the period of 2003–2012. (a, g) show the LIM3 averages, (b, h) the difference between PHC3 (Steele et al., 2001) and LIM3, (c, i) the difference between WOA13 (Boyer et al., 2013) and LIM3, (e, f, k, l) the corresponding differences for LIM2, and (d, j) show the differences between LIM3 and LIM2. WOA13 data are averaged over the years 2005–2013, while PHC3 data contain observations from 1900–1998, as in WOA98, plus all Arctic observations until 2004. Hatching indicates regions with statistically significant differences at the 5% level based on unequal variances t-test. As PHC3 does not include standard deviation and sample size, the statistical significance levels of LIM–PHC3 differences could not be estimated.

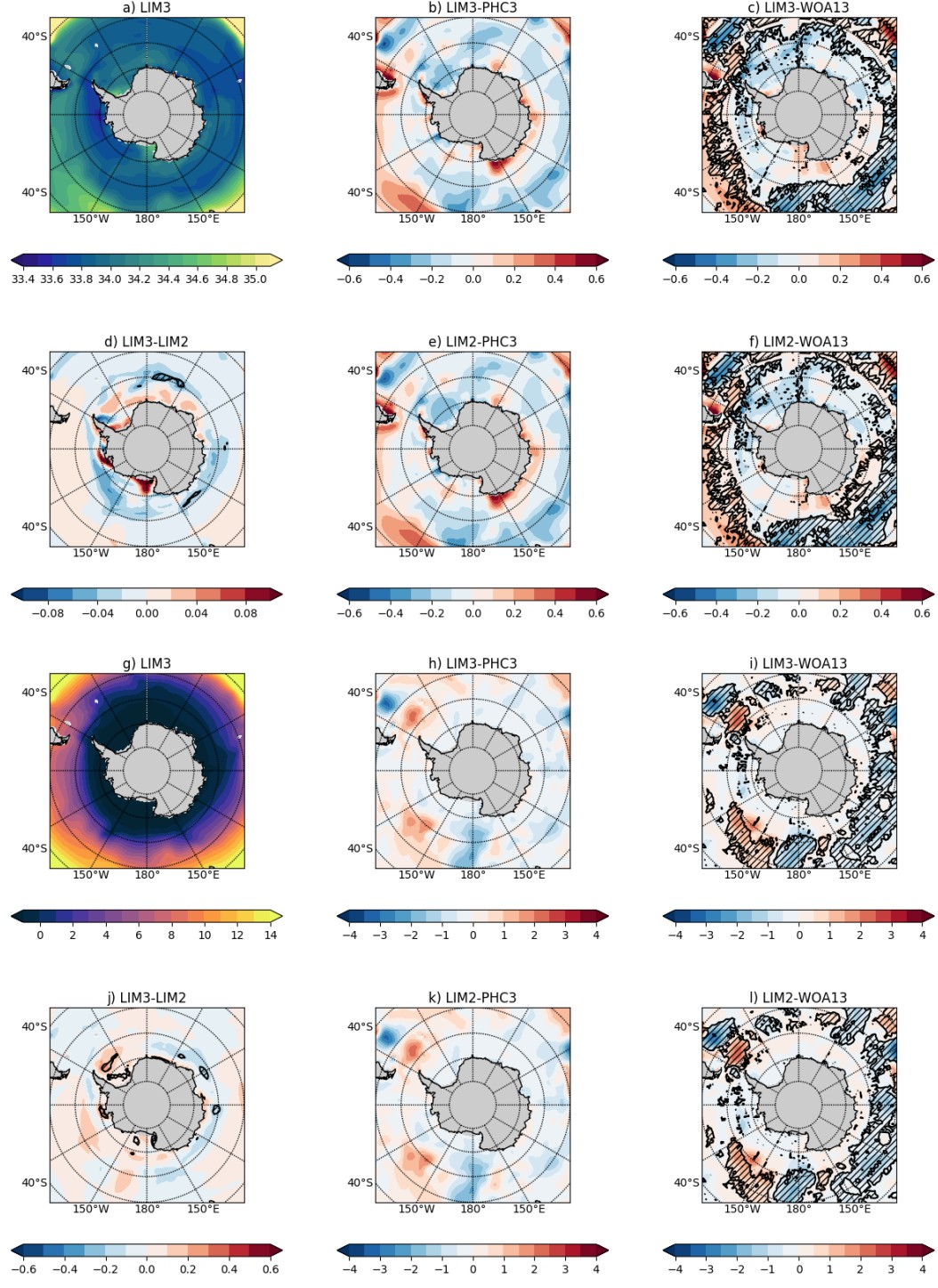

**Figure 6.** As Figure 5, but for the Southern Hemisphere.

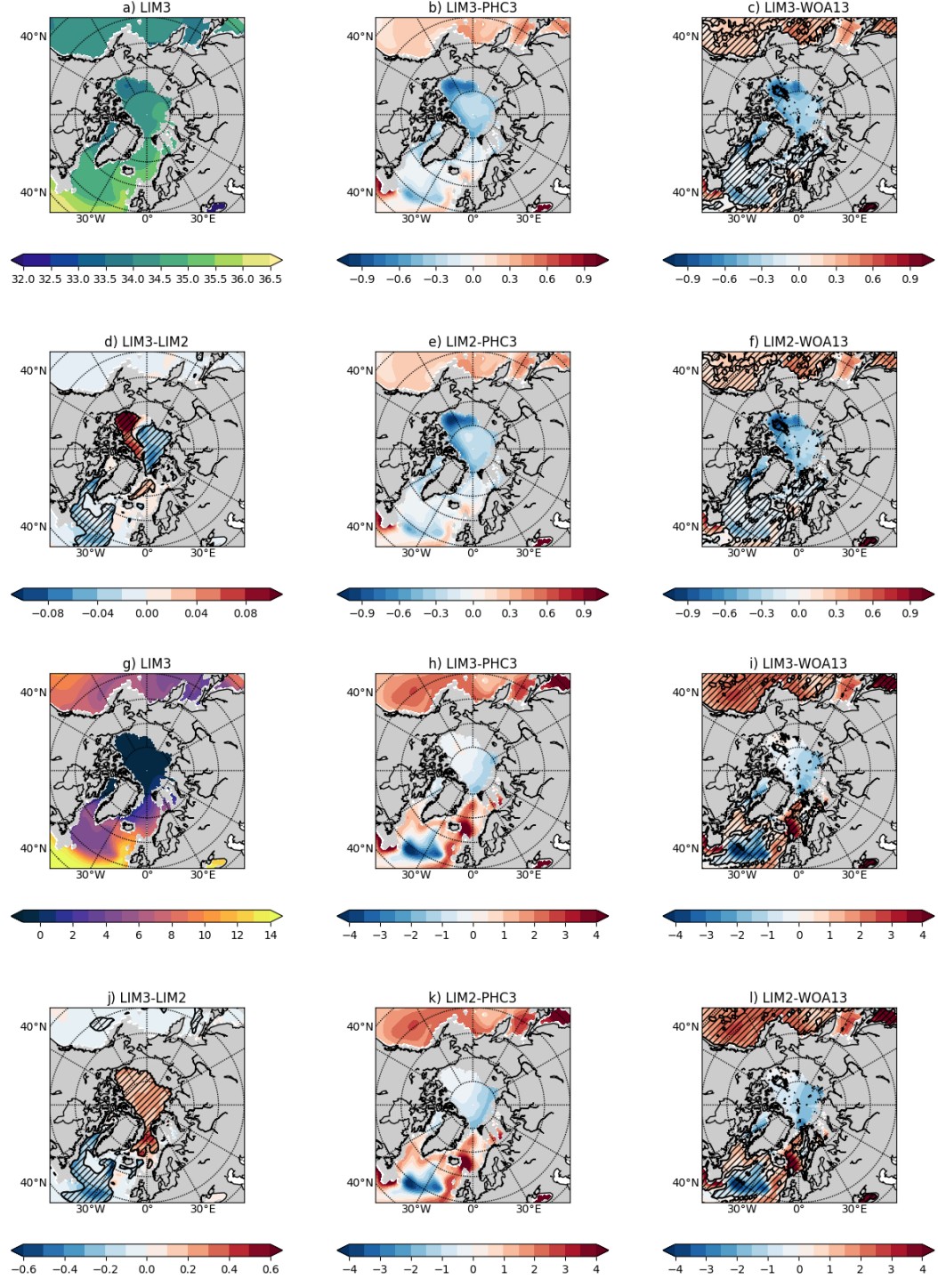

**Figure 7.** As Figure 5, but for the Arctic intermediate water (AIW) at 250 m.

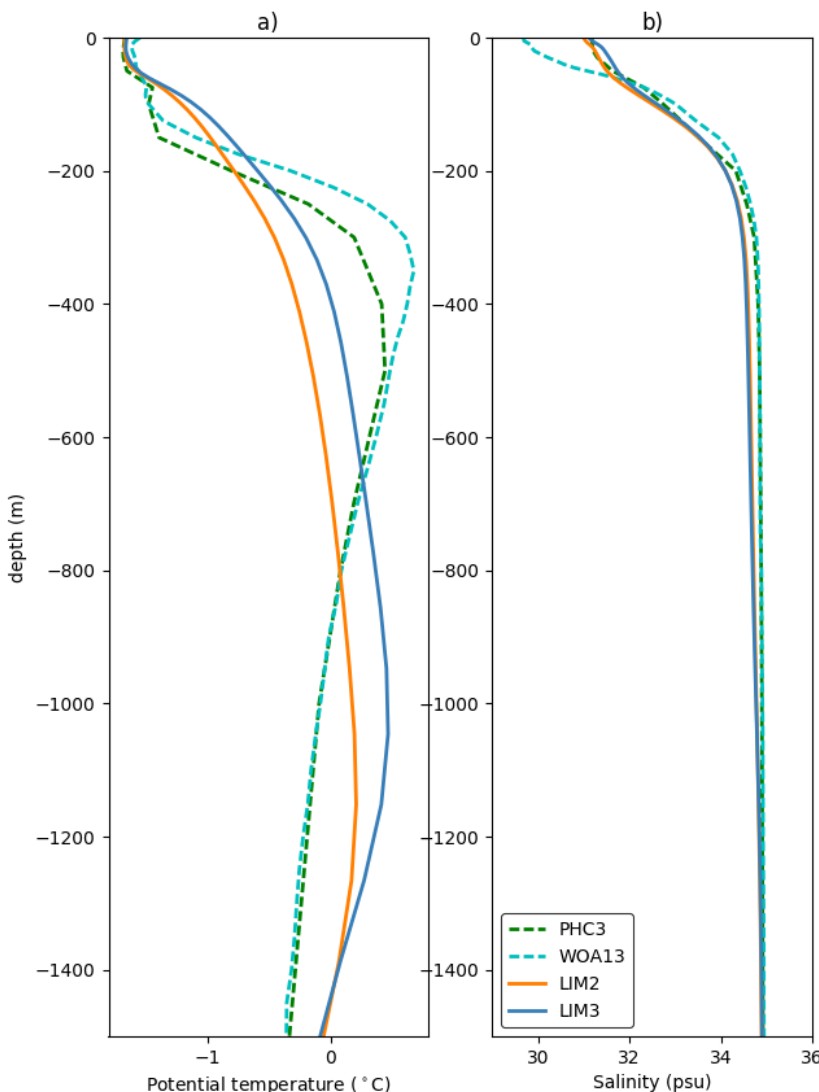

**Figure 8.** Vertical profiles of (a) potential temperature (°C) and (b) salinity (psu) at a grid point close to the North Pole (85.5°N, 140°W). Green dashed lines are based on the PHC3 climatology by Steele et al. (2001), cyan dashed lines are based on the WOA13 2005–2012 climatology by Boyer et al. (2013), orange lines show values from the NEMO-LIM2 simulation and blue lines from the NEMO-LIM3 simulation. NEMO-LIM profiles are averages over the years 2003–2012. Note that the PHC3 data were used to initialise two NEMO-LIM simulations, after which they largely lost their initially warm Atlantic Intermediate Water.

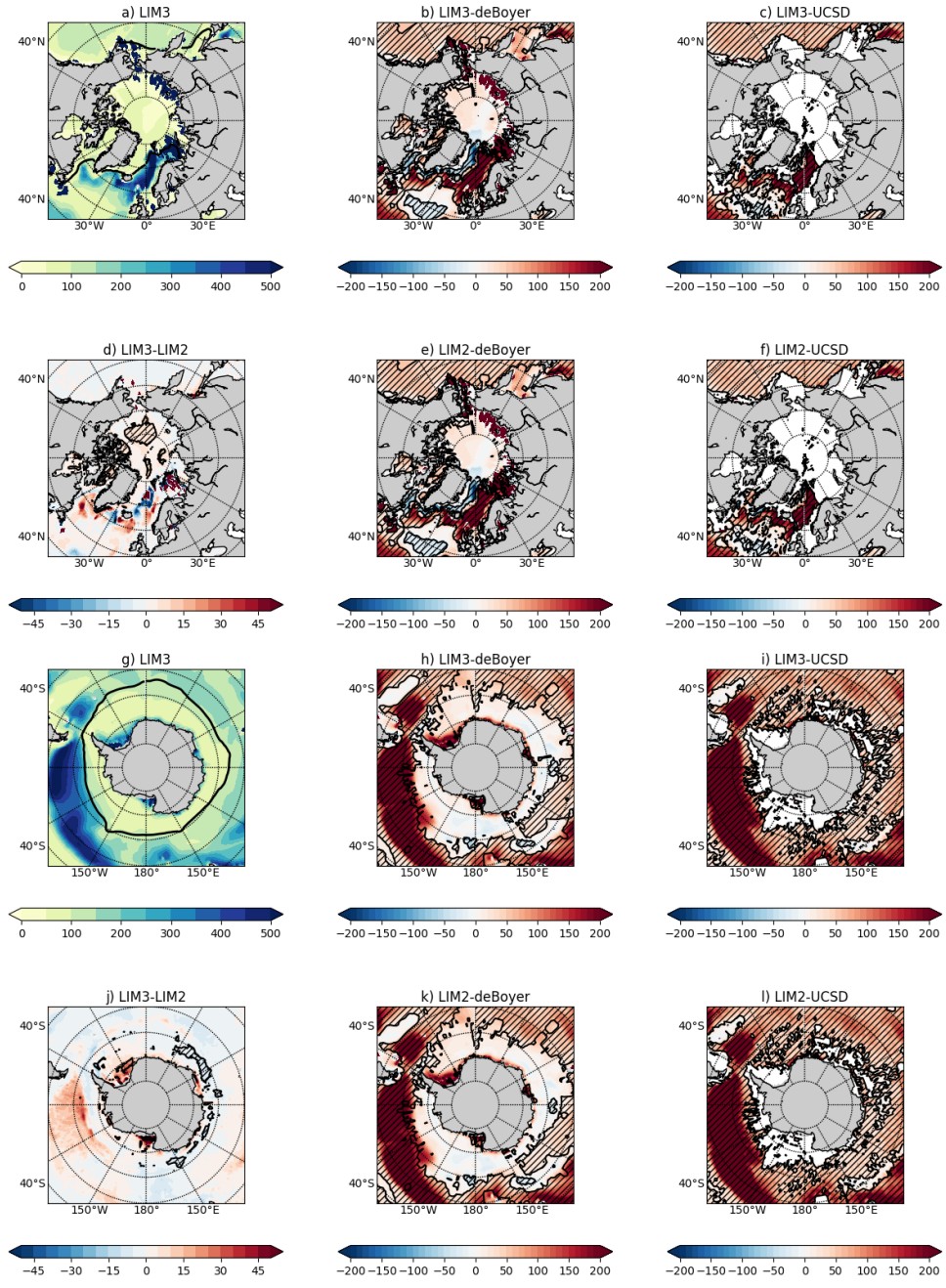

**Figure 9.** (a, g) mixed layer depths (in metres) as simulated by NEMO-LIM3, their departures from the observed climatologies of de Boyer Montégut et al. (2004) (b, h) and Holte et al. (2016) (c, i), the corresponding departures for NEMO-LIM2 (e, f, k, l) and differences between LIM3 and LIM2 (d, j). Top two row panels (a–f) represent March averages in the Northern Hemisphere (NH) and bottom two row panels (g–l) present September averages in the Southern Hemisphere (SH). Monthly averages were calculated from 2003–2012. Mixed layer depths are based on the potential density threshold value difference of 0.03 kg m$^{-3}$ from the density value at 10 m depth. In (a, g), thick black lines show the LIM3 sea-ice edge as the 15% sea-ice concentration isopleth. Hatching indicates regions with statistically significant differences at the 5% level based on unequal variances t-test.

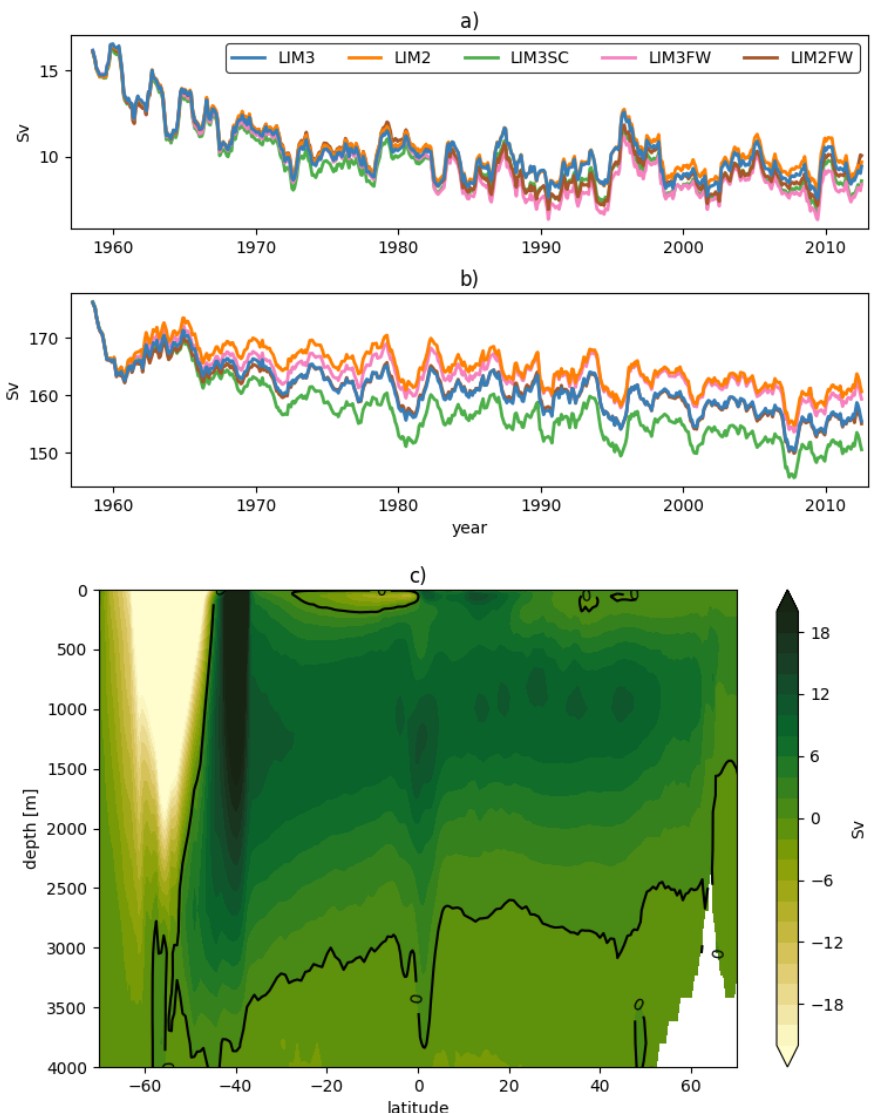

**Figure 10.** (a) Time series of the Atlantic Meridional Overturning Circulation (AMOC) in Sverdrups (Sv = $10^6$ m$^3$ s$^{-1}$) for NEMO-LIM3 simulation (blue line), NEMO-LIM2 simulation (orange line), NEMO-LIM3SC simulation (green line), NEMO-LIM3FW simulation (magenta line) and NEMO-LIM2FW simulation (brown line) integrated zonally across the Atlantic along the 50–53°N latitudinal band. In (b) the corresponding volume transport time series through the Drake Passage are shown. Time series in (a) and (b) represent 12 month running means. In (c), the Atlantic meridional transect of MOC (in Sv) for the NEMO-LIM3 simulation averaged over 2003–2012 as a function of depth and latitude is presented.

**Table 1.** NEMO3.6-LIM simulations analysed in this study.

| # | simulation name | number of sea-ice categories | snow thickness initialisation | sea-ice concentration initialisation | sea-ice strength | sea-ice salinity | sea-surface salinity restoring | freshwater budget correction |
|---|---|---|---|---|---|---|---|---|
| 1 | LIM3 | 5 | 0.3 m | 90% | $2\times10^4$ Nm$^{-1}$ | prognostic | true | annually |
| 2 | LIM2 | 1 | 0.5 m in NH, 0.1 m in SH | 90% in NH, 95% in SH | $1\times10^4$ Nm$^{-1}$ | constant, 4 ppm | true | annually |
| Sensitivity experiments: | | | | | | | | |
| 3 | LIM3SC | 1 | as in #1 | as in #1 | as in #1 | as in #2 | true | none |
| 4 | LIM3FW | 5 | as in #1 | as in #1 | as in #1 | as in #1 | false | none |
| 5 | LIM2FW | 1 | as in #2 | as in #2 | as in #2 | as in #2 | false | none |