# Peer review of "Comparing some aspects of the ocean hydrography, circulation and sea ice between NEMO3.6 LIM3 and LIM2"

_Geoscientific Model Development, 2016_

## Short Comment (SC1) · 22 Aug 2016

Dear authors,

In my role as Executive editor of GMD, I would like to bring to your attention our Editorial version 1.1:

http://www.geosci-model-dev.net/8/3487/2015/gmd-8-3487-2015.html

This highlights some requirements of papers published in GMD, which is also available on the GMD website in the 'Manuscript Types' section:

http://www.geoscientific-model-development.net/submission/manuscript_types.html

In particular, please note that for your paper, the following requirements have not been met in the Discussions paper:

- "The main paper must give the model name and version number (or other unique identifier) in the title."

- "If the model development relates to a single model then the model name and the version number must be included in the title of the paper. If the main intention of an article is to make a general (i.e. model independent) statement about the usefulness of a new development, but the usefulness is shown with the help of one specific model, the model name and version number must be stated in the title. The title could have a form such as, "Title outlining amazing generic advance: a case study with Model XXX (version Y)"."

Therefore I recomment to change the title of your article to something like: "On the influence of sea-ice physics in multi-decadal ocean-ice hindcasts: a NEMO 3.6 case study using the sea-ice schemes LIM2 and LIM3" in your revised submission to GMD.

Yours,

Astrid Kerkweg

---

## Referee Comment (RC1) · Anonymous Referee #1 · 21 Sep 2016

This article analyzes the effect of the new LIM3 sea ice model compared to the old LIM2 sea ice model in ocean stand-alone simulations with the new NEMO3.6 model. The results show an improvement of the sea ice representation but little effect on the rest of the ocean. Since the NEMO-ocean model is widely used in the climate modelling community and will also be the ocean component of a number of CMIP6 models, I find this comparison useful and worth to publish. The article is generally well written and organized. However, at a number of places, some more clarifications are needed and I find a few of the explanations for the differences between the model versions not entirely convincing.

Please find below a few more general comments, followed by a number of more specific points.

Main points: 1. Several times, the authors state that the main objective of this study is to evaluate ocean hydrography and circulation. The manuscript in its present form does not reflect this. While the comparison of sea ice representation between LIM2 and LIM3 is done in detail, the evaluation of the ocean circulation part is rather superficial, partly because differences between NEMO-LIM2 and NEMO-LIM3 are small in the ocean away from the ice. If the ambition of the authors really is to focus mainly on the evaluation of the ocean circulation and if this article should be the major reference for the performance of the new NEMO3.6 model, much more detailed analyses are needed. However, if the main idea is to specifically focus on the effect of LIM3 and LIM2 in NEMO3.6, I would suggest to state that this article should be on: "Sea ice representation and some aspects of the ocean hydrography and circulation". In this case not much additional analysis is needed.

2. It would help to add also subfigures of "LIM2-Obs" in the figures. It is often very difficult to really judge from LIM3-Obs and LIM3-LIM2, how much LIM3 really improved the result, especially if the colour scales for LIM3-OBS and LIM3-LIM2 are different.

3. Differences between NEMO3.6-LIM2 and NEMO3.6-LIM3 are often rather small and taken over a relatively short period (10-years). Thus, significances of the differences should be calculated and shown in the figures.

4. The impact of the ice model on mixing, deep water formation and ocean circulation will take place through salinity changes. However, the restoring in the model (+ the prescribed atmosphere that cannot feed back onto the ocean) might hide much of this effect. Thus, the experiments without freshwater adjustments are very important in order to analyze the impact of the ice-model on the ocean and results from these experiments should be discussed more in detail.

5. It should be considered to reformulate the abstract. It is not very clear, includes some, for the abstract, unnecessary information and could instead mention some more of the major results from this study.

Specific comments:

Abstract: 1. p1, l6: "Results of such analysis . . ..": I do not think this justification is needed in the abstract

2. p1, l8: Delete "while NEMO-LIM2 deviates more"

3. p1, l11: "skill sufficient for ocean-ice hindcasts that target oceanographic studies": unclear, make clearer or delete

4. P1, l17-20: Since coupling to the atmosphere is mentioned, the potential effect of ice variations/ trends on atmospheric circulation should be shortly discussed as well (e.g. Barnes 2013; Francis et al. 2009; Francis and Vavrus 2012; Garcia Serrano and Frankignoul 2014; Hopsch et al. 2012; Koenigk et al. 2016; Liptak and Strong 2014; Overland and Wang 2010; Petoukhov and Semenov 2010; Screen 2014; Yang and Christensen 2012, . . ..). One motivation to improve sea ice models is that this might have large consequences on atmospheric climate conditions as well.

5. P2, l2-3: See main point 1: The main focus of this study seems to be the effect on sea ice and not on ocean circulation, which is by far less intensive analyzed in this study.

6. P3, l12: I thought the minimum horizontal length is a bit smaller than 50km in the Arctic near the poles, e.g. around Greenland. Please check.

7. P4, l17: Please explain what is meant with a salinity restoring rate of -100mm/day. If this is a freshwater flux (global average?), this sounds very large.

8. P4, l30-P5, l2: I am confused about what tuning has been done for each of the versions? Here, it is stated that a specific and optimized tuning has been done for each of the versions. In the conclusions you state; "no specific tuning has been done". I agree that two optimized model versions should be compared. In this context, I wonder if really the same effort has been done to optimize NEMO3.6LIM2 as for optimizing NEMO3.6LIM3? My worry is that the LIM2-ice-parameters have been taken from an

older NEMO-LIM2 version and that the NEMO3.6-ocean parameters have been tuned with LIM3 and not with LIM2. Please describe in more detail how these two versions have been tuned and optimized.

9. P5, l4-10: I am not sure I really understand this: Are you saying that LIM2 with P* from LIM3 simulates much less ice volume but the same ice area than LIM2 with its standard P*? Why is this indicating "insignificant oceanic impacts"? Please clarify.

10. Experiments: A table listing the different simulations would be helpful

11. P5, l15: If LIM3 with 1 ice category is much better than LIM2 but physically closer to LIM2 than LIM3, what is the reason that LIM3-1IC is better? This feature is then obviously more important for an ice model than e.g. multiple ice thickness categories.

12. P6, l 24: Again, I do not have the impression that this study only shortly focuses on the sea ice and merely on sensitivity experiments and oceanographic analysis. Section 3 is the longest of all sections.

13. P6, l29, Figure1 b: I do not see 50% reduction in the East-Siberian Sea. Largest reduction seems to be between North Pole and northern Kara Sea. Please check.

14. Figure 1: It would be good to show the spatial distribution of LIM3MC as well.

15. Figure2: It is not really self-explanatory to call LIM3 with 1 single ice category "LIM3MC". This sounds like LIM3 with multi-ice categories. Maybe better LIM3SC.

16. P 7, l34: I do not think you can explain the LIM3 low summer ice extent by the ice-albedo-feedback. The ice-albedo feedback exists in reality. Maybe you can explain the low summer ice extent by too low ice albedo in LIM3 or a too strong effect due to unrealistic distribution of ice thicknesses (e.g too much thin ice). On the other hand, the annual cycle in LIM2 is even larger than in LIM3 (which is opposite to the NH). Does not this speak against an effect of the ice categories?

17. P8, l21: I think your numbers are wrong. PIOMAS shows values of about $4\text{-}8\times10^3$

km3 in September and 20-25x103 km3 in early spring in the last decade. Please check your values.

18. P8, l26: LIM2 shows a stronger negative trend as LIM3? This is thus opposite to the ice area trends?

19. P9, l18: Here you state that the sea ice albedo feedback is less important in the SH. I agree but this is in contradiction to the argumentation before that the stronger sea ice albedo feedback in LIM3 explains the LIM3-low summer ice extents (see point 16).

20. Figure 3: The displacement of the Beaufort Gyre in LIM3 seems to agree with the positive ice bias and the general tendency of LIM to have the thickest ice displaced/ extended towards the Northern American coast.

21. P10, l12/13: I do not understand: "...at regions." Sentence uncomplete?

22. P 10, L25: Yes, but on the other hand LIM3MC with constant salinity does perform quite well. Without simulations that separately analyze the impact of the different new features in LIM3, it is very speculative to argue that the prognostic sea ice salinity improved ice volume and area. The results from section 4.1 do not support the conclusion that prognostic salinity is strongly improving the ice volume/ extent. Please rethink this statement here.

23. P12, l 22ff: 54 years are rather short from an ocean circulation point of view. A systematic decrease by 1-2 Sv in the AMOC might lead to larger effects on the ice after longer periods. Furthermore, I think the statement that "if problems appear, they are related to the coupling .." might be misinterpreted. In the uncoupled ocean-ice model, the atmosphere cannot feedback to the ocean. Thus, the effect of salinity/ freshwater changes on the ice is probably not very large. However, in a coupled system, small changes in the freshwater balance or related changes in AMOC and SST-pattern could lead to strong effects in the atmosphere, which in turn might strongly affect the ocean currents, ocean heat transports and ocean-ice coupling as well. Thus, changing the

freshwater balance in the ocean could create important issues in a coupled model and could be the reason for performance issues in the coupled model. Please reformulate to avoid misunderstanding.

24. From Fig. 5 it seems that more freshwater is transported in the East Greenland Current to the south and then further into the Labrador Sea. This might be related to the fact that there is more ice in the Greenland Sea in LIM2, which leads to a more constraint freshwater transport in the EGC in LIM2 than in LIM3. However, you relate the lower salinity in LIM2 in the Labrador Sea to more local melting. To decrease salinity this would also need a net transport of sea ice into the Labrador Sea because stronger local ice growth and ice melt would not decrease the SSS. Did you analyze net ice-growth rates in LIM3 and LIM2?

25. Figure 5 g: I am a bit surprised over the cold bias in NEMO3.6-LIM3 in the Fram Strait-Svalbard area. Is this related to too high ice velocities and too much ice in this area? For September, Figure1 b does not seem to indicate too much ice but maybe in the rest of the year. Please add a sentence on this.

26. Figures 5/6 and salinity discussion: Given the fact that SSS is quite strongly restored in NEMO-LIM2 and NEMO-LIM3, can we really conclude from the small differences in SSS that the sea ice model has a small impact on the salinity distribution? Are the SSS-differences between LIM2 and LIM3 as small in the experiments without freshwater adjustments?

27. P15, l1,2: Why is a larger Atlantic warm water inflow associated with a smaller AMOC? There is some discussion in the community how strong the AMOC is linked to the ocean heat transport into the Arctic but most studies suggest that an increased AMOC leads to increased transports of Atlantic water masses into the Arctic.

28. Fig. 9: Are you sure that the 15% ice edge is at the right place and really the observed ice edge? It goes very far to the south and the east in the Greenland Sea and also in the Labrador Sea. Please check.

29. LIM3-Ref also indicates deep convection in the Greenland Sea far inside the ice area. Further NEMO does not show any deep convection in the Labrador Sea but the climatology does not either. Results from ARGO-floats, which cover the time period 2000-2015 (Holte et al. 2010; http://mixedlayer.ucsd.edu/) show deep convection in the Labrador Sea and might be more reliable than the climatology used in this study.

30. P16 AMOC: The observational based estimates should be mentioned: e.g. RAPID: 16.9 Sv (at 26.5N), Ganachaud (2003) and Lumpkin and Speer (2007): 18.5 Sv (at 24N) and 16.5 Sv at 48N (Ganachaud). There are many things well simulated in NEMO, but unfortunately not the AMOC ….

31. P17, l1-3: Again, the SSS-restoring might hide differences between LIM3 and LIM2: Is the AMOC-difference between LIM3 and LIM2 the same in the experiments without freshwater adjustments?

32. P 17, l20: I would delete "briefly". As stated before: this comparison is - if not the main - but an important part of this study. 3 3. P17, l22 and l29-30: The conclusions on the sea ice albedo puzzle me: First you argue that the better representation of the sea ice albedo feedback is the main improvement and then you argue that the model is stable to changes in summer albedo. The first is also related to the different thickness classes but is it really a sign for realism if the model is insensitive to the change of summer albedo? How much is the summer albedo changing with the new sea ice albedo scheme? Maybe, the difference is small?

34. P 18, l3: I think it is a bit overstated to say that you evaluated the oceanic transports across major transects of the world ocean. You only looked at the AMOC and the Drake Passage. You do not show any results from ocean heat transports in the different oceans or transports into the Arctic or overflows (Denmark Strait, Iceland-Faroe-Scotland).

Typings, etc.

p1, l7"while NEMO-LIM2 deviates more". Could be deleted, if LIM3 agrees better than it is clear that LIM2 deviates more.

P13, l6/ l7: "melted freshwater" sounds weird. Better: "freshwater from melted sea ice"

P 14, l10: delete one "be"

P17, l13: A set . . . "was" performed.

P 17, l13: ". . . in the global ORCA1 grid": Add a "configuration" or "using the global ORCA1 grid.

---

## Referee Comment (RC2) · Anonymous Referee #2 · 2 Nov 2016

**General comments**

The paper presents an evaluation of the updates to the LIM sea ice model, within the NEMO ocean model, version 3.6. A comparison is made between the previous configuration, LIM2, and the new one, LIM3. Some sensitivity runs are performed, to allow differences between LIM2 and LIM3 to be attributed to particular configuration changes. The model output is also evaluated against appropriate observational datasets. Only the sea ice configuration, and not that of the ocean, has changed between the two model versions, but the authors carry out a detailed study of the impact on the ocean as well as the sea ice. The paper is well-structured and methodical. It is generally scientifically robust, although I have some specific concerns which I will outline below. The paper is well-written, and the standard of English is generally good. I recommend

publication after the issues outlined below have been addressed.

**Specific comments**

- Five model configurations are analysed in the paper – LIM2, LIM3, LIM3MC, and the two simulations to test sensitivity to the freshwater adjustments. The configurations are described in detail in Sections 2.4, 4.1 and 4.2. However, I think that a useful complement to this would be a table summarising the settings in each configuration. That way, the reader could refer back easily to the table while studying the results.

- In several figures, the authors use both green and red. This should be avoided if possible, as readers who are colour-blind will have difficulty in distinguishing between these colours. The "rainbow" colour scheme used in Figure 3 should be avoided for the same reason if possible.

- The atmospheric forcing dataset used by the authors is based on ERA-40 before 2001, and ERA-Interim thereafter. Several authors have cast doubt on the reliability of ERA-40 in the polar regions, due to the sparsity of observations there. For example, Screen Simmonds (2011) noted a discontinuity in Arctic temperature in 1997, leading to a significantly exaggerated warming in the mid-to-lower troposphere. While the main period considered by the authors, 2003-2012, is wholly covered by the ERA-Interim-based forcing, the period immediately before could be affected by the inaccuracies in ERA-40, and there could be some residual impact in 2003-2012 due to the "memory" of the ocean. In addition, the authors do sometimes make use of the model output from before 2003 (for example, when discussing the trends in ice extent since 1979 as presented in Figure 2b). They should therefore reflect in the paper on the possible impact on the results of any inaccuracies in the ERA-40 forcing data.

- In Section 3.1, where the authors refer to the trends in SH ice extent as "statistically significantly increasing", they should specify the level of significance and the method used to assess it. In a number of other places in the sea ice sections, the authors refer to results as "significant". For example, in Section 3.2, the interannual variations in annual mean NH ice volume are "significant". In the same section, the LIM3 SH ice volume is said to have a "significant positive trend", the trend in GIOMAS ice volume is "significantly negative", and LIM2 has "no significant trend". There are further examples in the discussion of the sensitivity runs in Section 4. Do the authors mean "statistically significant"? And if so, how was this assessed and what is the level of significance? This should be specified in the paper.

- In the discussion of the impacts on the ocean (Section 5), there is no discussion of statistical significance of the results. In a number of cases, I wondered if the inter-model differences, and the differences with respect to observations, were significant with respect to the interannual variability, especially where the differences were small. For example, in Section 5.6, the authors refer to "small temperature and salinity differences" between the LIM simulations. And in Section 5.8, they refer to the fact that the AMOC in LIM2 is up to 0.4 Sv stronger than that in LIM3. Are these differences statistically significant above the interannual variability?

- In most cases, the authors restrict their analysis to the last decade of the 54-year simulations. Given that the model starts from rest and will take some time to spin up, this is the correct approach. However, restricting the period of study to 10 years means that multidecadal variability will not be picked up in the analysis. To what extent are the differences between LIM2 and LIM3, or the differences between the LIM configurations and observations, dependent on factors that may be subject to multidecadal variability? I appreciate it may be difficult to answer this question with the results available, but I think the authors should at least

mention it in the discussion of their results.

- In Section 3.2 (page 10, lines 12-13), the authors state "Close to the ice edge, LIM3 has a smaller ice extent and a lower ice concentration at regions". I'm not sure what is meant by "at regions". Does it mean "in certain locations"? If so, then please change it.

- In the conclusions, the authors refer to the new sea ice albedo scheme implemented in LIM3 in April 2016. They state "Our preliminary tests on this new scheme demonstrate the robustness of LIM3, as its sea-ice distribution appears almost insensitive to changes in summer albedo". I find this statement surprising, as several studies have shown that sea ice simulations are indeed extremely sensitive to albedo in summer (see, e.g., Kim et al., 2006, and Rae et al., 2014). While use of a reliable atmospheric forcing dataset should reduce the need for tuning via albedo adjustments (see, e.g., Hunke, 2010), I would still expect the sea ice simulation to be sensitive to albedo changes. Perhaps I have misunderstood what the authors mean by this statement, but I would appreciate some clarification.

- In the figures, the results are generally presented as LIM3-LIM2, or LIM3-obs. This makes sense, as the authors are presenting the first study of a simulation with the new LIM3 configuration, and comparing it with the previous configuration (LIM2) and observations. However, in the text the authors often discuss the results in reverse. For example, on page 16, line 3, when referring to Figure 9e: "In the Southern Ocean, the observed mixed layer is shallower than the LIM3 mixed layer...". This wording seems a bit strange and "back-to-front" to me, as the results in the figure are presented as LIM3 minus obs (and anyway the purpose of the paper is to assess LIM3 against observations, not the other way round). I would prefer to see the wording in the text reflect the way the results are presented in the figures ("In the Southern Ocean, the LIM3 mixed layer is deeper

than observed"), unless there is a good reason to do otherwise. There are several other examples of this type of "back-to-front" wording throughout the paper to which similar remarks apply.

**Technical corrections**

- Where the authors mention results without including figures (e.g. in Section 3.2, although there are other places where this occurs), it would be useful for the reader if they stated "not shown here".

- Page 7, line 5: "Figures 2a and b" should be "Figures 2a and c".

- Page 7, line 14: "minimum extends" should be "minimum extents".

- Page 9, line 15: "a lesser significance as in the..." should be "a lesser significance than in the...".

- Page 14, line 4: "smaller between LIM3 and WOA13 than between PHC3" should be "smaller between LIM3 and WOA13 than between LIM3 and PHC3"

- Page 14, line 21: "around the East Antarctica" should be "around the East Antarctic" or "around East Antarctica" (without "the").

- Page 14, line 30: "loose heat to the water masses" should be "lose heat to the water masses".

- Page 15, line 2: "might be associated to" should be "might be associated with".

- Page 15, line 33: "This is at least partly due to cold, non-responsive and prescribed winter atmosphere..." should be "This is at least partly due to the cold, non-responsive prescribed winter atmosphere..." (add "the", delete "and").

- Page 16, line 10: "and one where denser LIM2 surface entrain deeper" should be "and one where the denser LIM2 surface entrains more deeply".

- Page 16, line 11: "with most distinct ones visible" should be "with the most distinct ones visible".

- Page 16, line 18: "Danabasoglu et al. (2014) assessed the mean AMOC of eighteen ocean-ice models forced by prescribed atmospheric forcing from 1948–2007 and run five repetitive forcing cycles...": Inconsistent tenses (past/present). "run" should be "ran".

- Page 16, line 20: "Here, results of..." should be "Here, the results of...".

**References**

Hunke, E.C., 2010. Thickness sensitivities in the CICE sea ice model. Ocean Modell. 34, 137–149. http://dx.doi.org/10.1016/j.ocemod.2010.05.004.

Kim, J.G., Hunke, E.C., and Lipscomb, W.H., 2006. Sensitivity analysis and parameter tuning scheme for global sea-ice modeling. Ocean Modell. 14, 61–80. http://dx.doi.org/10.1016/j.ocemod.2006.03.003.

Rae, J.G.L., Hewitt, H.T., Keen, A.B., Ridley, J.K. Edwards, J.M., and Harris, C.M., 2014. A sensitivity study of the sea ice simulation in the global coupled climate model, HadGEM3. Ocean Modell. 74, 60—76. http://dx.doi.org/10.1016/j.ocemod.2013.12.003.

Screen, J.A., and Simmonds, I., 2011. Erroneous Arctic Temperature Trends in the ERA-40 Reanalysis: A Closer Look. J. Climate, 24. 2620–2627. doi: 10.1175/2010JCLI4054.1.

---

## Author Comment (AC1) · 22 Dec 2016

Thank you very much for pointing these journal guidelines out. Following them and Reviewer #1 suggestions we have changed the title of our article to: "Comparing some aspects of the ocean hydrography, circulation and sea ice between NEMO3.6 LIM2 and LIM3". We think this new title well describes the manuscript content and matches the GMD journal requirements.

---

## Author Comment (AC2) · 22 Dec 2016

**General comments**

*The paper presents an evaluation of the updates to the LIM sea ice model, within the NEMO ocean model, version 3.6. A comparison is made between the previous configuration, LIM2, and the new one, LIM3. Some sensitivity runs are performed, to allow differences between LIM2 and LIM3 to be attributed to particular configuration changes. The model output is also evaluated against appropriate observational datasets. Only the sea ice configuration, and not that of the ocean, has changed between the two model versions, but the authors carry out a detailed study of the impact on the ocean as well as the sea ice. The paper is well-structured and methodical. It*

*is generally scientifically robust, although I have some specific concerns which I will outline below. The paper is well-written, and the standard of English is generally good. I recommend publication after the issues outlined below have been addressed.*

Author response: We thank the reviewer for her/his supportive comments and excellent suggestions that significantly improve the manuscript. We followed all the points she/he rose. Please find our detailed responses below.

**Specific comments**

- *Five model configurations are analysed in the paper – LIM2, LIM3, LIM3MC, and the two simulations to test sensitivity to the freshwater adjustments. The configurations are described in detail in Sections 2.4, 4.1 and 4.2. However, I think that a useful complement to this would be a table summarising the settings in each configuration. That way, the reader could refer back easily to the table while studying the results.* Author response: A good suggestion, thanks. We added Table 1 which lists the simulations used for the study and their main characteristics.

- *In several figures, the authors use both green and red. This should be avoided if possible, as readers who are colour-blind will have difficulty in distinguishing between these colours. The "rainbow" colour scheme used in Figure 3 should be avoided for the same reason if possible.* Author response: We followed your advice and use now different line colours. Furthermore, we do not use the "rainbow" colour scheme in Figure 3 any more.

- *The atmospheric forcing dataset used by the authors is based on ERA-40 before 2001, and ERA-Interim thereafter. Several authors have cast doubt on the reliability of ERA-40 in the polar regions, due to the sparsity of observations*

*there. For example, Screen Simmonds (2011) noted a discontinuity in Arctic temperature in 1997, leading to a significantly exaggerated warming in the mid-to-lower troposphere. While the main period considered by the authors, 2003-2012, is wholly covered by the ERA-Interim-based forcing, the period immediately before could be affected by the inaccuracies in ERA-40, and there could be some residual impact in 2003-2012 due to the "memory" of the ocean. In addition, the authors do sometimes make use of the model output from before 2003 (for example, when discussing the trends in ice extent since 1979 as presented in Figure 2b). They should therefore reflect in the paper on the possible impact on the results of any inaccuracies in the ERA-40 forcing data.* Author response: This is a mistake from our side which has now been corrected in the text. After double checking the description of DFS 5.2 forcing data, see Dussin and Barnier (2014, https://www.drakkar-ocean.eu/forcing-the-ocean/the-making-of-the-drakkar-forcing-set-dfs5), we realise that the DRAKKAR 5.2 data are based on ERA-Intering since 1979, not since 2001, which is the case of earlier versions of DRAKKAR data. Before 1979, the DRAKKAR 5.2 data are based on a combination of ERA-40 and ERA-Interim climatology. Hence, the inaccuracies of ERA-40 have a minor impact on the main study periods. We thank the reviewer for leading us to find our mistake.

- *In Section 3.1, where the authors refer to the trends in SH ice extent as "statistically significantly increasing", they should specify the level of significance and the method used to assess it. In a number of other places in the sea ice sections, the authors refer to results as "significant". For example, in Section 3.2, the interannual variations in annual mean NH ice volume are "significant". In the same section, the LIM3 SH ice volume is said to have a "significant positive trend", the trend in GIOMAS ice volume is "significantly negative", and LIM2 has "no significant trend". There are further examples in the discussion of the sensitivity runs in Section 4. Do the authors mean "statistically significant"? And if*

*so, how was this assessed and what is the level of significance? This should be specified in the paper.* Author response: When referring to the sea-ice trends and differences we mean the statistical significance at the 5% level. We reformulated the text and now explicitly specify the level of statistical significance and the methods to assess it. In other occasions, where we used the word 'significant', or its derivatives, in a descriptive sense and did not mean 'statistically significant', we replaced 'significant' with other words, such as 'remarkable', 'clear' or 'apparent'. These expressions refer to differences or findings which appear physically unambiguous.

- *In the discussion of the impacts on the ocean (Section 5), there is no discussion of statistical significance of the results. In a number of cases, I wondered if the inter-model differences, and the differences with respect to observations, were significant with respect to the interannual variability, especially where the differences were small. For example, in Section 5.6, the authors refer to "small temperature and salinity differences" between the LIM simulations. And in Section 5.8, they refer to the fact that the AMOC in LIM2 is up to 0.4 Sv stronger than that in LIM3. Are these differences statistically significant above the interannual variability?* Author response: This is a good point, thanks. We calculated significance levels between LIM3 and LIM2 variables and between LIM simulations and observational data whenever possible. Following these calculations we now indicate significant differences at the 5% level in figures representing geographical climatological distributions. Related discussion has been added to the text.

- *In most cases, the authors restrict their analysis to the last decade of the 54-year simulations. Given that the model starts from rest and will take some time to spin up, this is the correct approach. However, restricting the period of study to 10 years means that multidecadal variability will not be picked up in the analysis. To what extent are the differences between LIM2 and LIM3, or the differences between the LIM configurations and observations, dependent on factors that may*

[Figure]

*be subject to multidecadal variability? I appreciate it may be difficult to answer this question with the results available, but I think the authors should at least mention it in the discussion of their results.* Author response: This is a good discussion point which we added to Conclusions section. Judging from Figure 1 multi-decadal sea-ice extent time series, it seems sensible to assume that LIM2–LIM3 differences are not very sensitive to interdecadal variability during the last few decades of simulations. This is because the mutual annual minimum, mean and maximum sea-ice extent differences remain systematic between the simulations. We can assume that upper ocean differences behave like the sea-ice ones, while deeper in the ocean the differences remain quite small.

- *In Section 3.2 (page 10, lines 12-13), the authors state "Close to the ice edge, LIM3 has a smaller ice extent and a lower ice concentration at regions". I'm not sure what is meant by "at regions". Does it mean "in certain locations"? If so, then please change it.* Author response: Yes, this sentence has a bad wording. We changed it to: "LIM3 has a smaller ice extent and a lower ice concentration close to the ice edge (not shown here).

- *In the conclusions, the authors refer to the new sea ice albedo scheme implemented in LIM3 in April 2016. They state "Our preliminary tests on this new scheme demonstrate the robustness of LIM3, as its sea-ice distribution appears almost insensitive to changes in summer albedo". I find this statement surprising, as several studies have shown that sea ice simulations are indeed extremely sensitive to albedo in summer (see, e.g., Kim et al., 2006, and Rae et al., 2014). While use of a reliable atmospheric forcing dataset should reduce the need for tuning via albedo adjustments (see, e.g., Hunke, 2010), I would still expect the sea ice simulation to be sensitive to albedo changes. Perhaps I have misunderstood what the authors mean by this statement, but I would appreciate some clarification.* Author response: Here, our wording appears wrong, as this is not what we meant. Thank you for pointing this out. We rewrote the sentence to be:

[Figure]

"This new sea-ice albedo scheme, with better transitions between the different ice types, slightly modifies the surface albedo compared to the old scheme and affects the model behaviour to a limited extent only."

- *In the figures, the results are generally presented as LIM3-LIM2, or LIM3-obs. This makes sense, as the authors are presenting the first study of a simulation with the new LIM3 configuration, and comparing it with the previous configuration (LIM2) and observations. However, in the text the authors often discuss the results in reverse. For example, on page 16, line 3, when referring to Figure 9e: "In the Southern Ocean, the observed mixed layer is shallower than the LIM3 mixed layer...". This wording seems a bit strange and "back-to-front" to me, as the results in the figure are presented as LIM3 minus obs (and anyway the purpose of the paper is to assess LIM3 against observations, not the other way round). I would prefer to see the wording in the text reflect the way the results are presented in the figures ("In the Southern Ocean, the LIM3 mixed layer is deeper than observed"), unless there is a good reason to do otherwise. There are several other examples of this type of "back-to-front" wording throughout the paper to which similar remarks apply.* Author response: A good point, thanks. The "back-to-front" wording carries from the stage when we plotted obs-LIM3 and we did not realise to change the wording. This has now been done and we do not spot "back-to-front" wording any more.

**Technical corrections**

- *Where the authors mention results without including figures (e.g. in Section 3.2, although there are other places where this occurs), it would be useful for the reader if they stated "not shown here".* Author response: A good point, we now state "not shown here" where we mention results without figures.

[Figure]

- *Page 7, line 5: "Figures 2a and b" should be "Figures 2a and c".* Author response: Well spotted. We changed the text accordingly.

- *Page 7, line 14: "minimum extends" should be "minimum extents".* Author response: Corrected.

- *Page 9, line 15: "a lesser significance as in the..." should be "a lesser significance than in the...".* Author response: Amended as suggested.

- *Page 14, line 4: "smaller between LIM3 and WOA13 than between PHC3" should be "smaller between LIM3 and WOA13 than between LIM3 and PHC3"* Author response: This is right. We modified the sentence accordingly.

- *Page 14, line 21: "around the East Antarctica" should be "around the East Antarctic" or "around East Antarctica" (without "the").* Author response: Changed to "around the East Antarctic".

- *Page 14, line 30: "loose heat to the water masses" should be "lose heat to the water masses".* Author response: Corrected.

- *Page 15, line 2: "might be associated to" should be "might be associated with".* Author response: Changed "associated to" to "associated with".

- *Page 15, line 33: "This is at least partly due to cold, non-responsive and prescribed winter atmosphere..." should be "This is at least partly due to the cold, non-responsive prescribed winter atmosphere..." (add "the", delete "and").* Author response: deleted.

- *Page 16, line 10: "and one where denser LIM2 surface entrain deeper" should be "and one where the denser LIM2 surface entrains more deeply".* Author response: Modified as suggested.

[Figure]

- *Page 16, line 11: "with most distinct ones visible" should be "with the most distinct ones visible".* Author response: Done.

- *Page 16, line 18: "Danabasoglu et al. (2014) assessed the mean AMOC of eighteen ocean-ice models forced by prescribed atmospheric forcing from 1948–2007 and run five repetitive forcing cycles...": Inconsistent tenses (past/present). "run" should be "ran".* Author response: Corrected.

- *Page 16, line 20: "Here, results of..." should be "Here, the results of...".* Author response: Changed.

---

## Author Comment (AC3) · 22 Dec 2016

*This article analyzes the effect of the new LIM3 sea ice model compared to the old LIM2 sea ice model in ocean stand-alone simulations with the new NEMO3.6 model. The results show an improvement of the sea ice representation but little effect on the rest of the ocean. Since the NEMO-ocean model is widely used in the climate modelling community and will also be the ocean component of a number of CMIP6 models, I find this comparison useful and worth to publish. The article is generally well written and organized. However, at a number of places, some more clarifications are needed and I find a few of the explanations for the differences between the model versions not entirely convincing.*

[Figure]

Author response: We thank the reviewer for carefully reading the manuscript and for her/his constructive suggestions that significantly improved the manuscript. Please find below our responses to the reviewer's general comments and specific points.

**Main points:**

*1. Several times, the authors state that the main objective of this study is to evaluate ocean hydrography and circulation. The manuscript in its present form does not reflect this. While the comparison of sea ice representation between LIM2 and LIM3 is done in detail, the evaluation of the ocean circulation part is rather superficial, partly because differences between NEMO-LIM2 and NEMO-LIM3 are small in the ocean away from the ice. If the ambition of the authors really is to focus mainly on the evaluation of the ocean circulation and if this article should be the major reference for the performance of the new NEMO3.6 model, much more detailed analyses are needed. However, if the main idea is to specifically focus on the effect of LIM3 and LIM2 in NEMO3.6, I would suggest to state that this article should be on: "Sea ice representation and some aspects of the ocean hydrography and circulation". In this case not much additional analysis is needed.*

Author response: Thank you very much for this suggestion. Yes, our main idea is to focus on the effect of LIM3 and LIM2 in NEMO3.6. Hence, we decided to follow your suggestion and changed the title of our article to: "Comparing some aspects of the ocean hydrography, circulation and sea ice between NEMO3.6 LIM2 and LIM3". We think this new title well describes the manuscript content and matches the GMD journal requirements, as pointed out by the Editor.

We would also like to make a point that the majority of oceanic diagnostics we carried

out, such as hydrography and transports, were excluded from the manuscript because they showed very small differences between LIM2 and LIM3, or the differences had similar characteristics than what the oceanic diagnostics included in the manuscript reveal. We think this is due to the fact that the largest impacts of the sea-ice model are concentrated to the upper ocean. Therefore we think that is fair to say that the oceanic analysis has had a large focus, in addition to sea-ice, although only a small part of it ended up in the present version of the manuscript due to the reasons mentioned.

*2. It would help to add also subfigures of "LIM2-Obs" in the figures. It is often very difficult to really judge from LIM3-Obs and LIM3-LIM2, how much LIM3 really improved the result, especially if the colour scales for LIM3-OBS and LIM3-LIM2 are different.*

Author response: A good point. We added "LIM2-Obs" panels to the figures.

*3. Differences between NEMO3.6-LIM2 and NEMO3.6-LIM3 are often rather small and taken over a relatively short period (10-years). Thus, significances of the differences should be calculated and shown in the figures.*

Author response: We agree. We used the t-test to estimate the 5% significances levels for average LIM3-LIM2 differences and hatched the areas of statistically significant differences in the figures.

*4. The impact of the ice model on mixing, deep water formation and ocean circulation will take place through salinity changes. However, the restoring in the model (+ the prescribed atmosphere that cannot feed back onto the ocean) might hide much of this effect. Thus, the experiments without freshwater adjustments are very important in order to analyze the impact of the ice-model on the ocean and results from these*

*experiments should be discussed more in detail.*

Author response: This is true and we concur. We have added panels to the figures and expanded the discussion on the experiments without freshwater adjustments. Our main finding is that the LIM3–LIM2 differences are smaller than LIM3FW–LIM2FW differences, in particular in the upper ocean. However, the difference patterns are remarkably similar.

*5. It should be considered to reformulate the abstract. It is not very clear, includes some, for the abstract, unnecessary information and could instead mention some more of the major results from this study.*

Author response: The abstract has been reformulated. We hope it is now more clear with necessary information and major results.

**Specific comments:**

*Abstract: 1. p1, l6: "Results of such analysis . . ..": I do not think this justification is needed in the abstract*

Author response: You are right. We have removed the sentence.

*2. p1, l8: Delete "while NEMO-LIM2 deviates more"*

Author response: Deleted.

*3. p1, l11: "skill sufficient for ocean-ice hindcasts that target oceanographic studies": unclear, make clearer or delete*

Author response: We clarified this sentence and state now that "... produced sea ice with a realism comparable to that of LIM2."

*4. P1, l17-20: Since coupling to the atmosphere is mentioned, the potential effect of ice variations/ trends on atmospheric circulation should be shortly discussed as well (e.g. Barnes 2013; Francis et al. 2009; Francis and Vavrus 2012; Garcia Serrano and Frankignoul 2014; Hopsch et al. 2012; Koenigk et al. 2016; Liptak and Strong 2014; Overland and Wang 2010; Petoukhov and Semenov 2010; Screen 2014; Yang and Christensen 2012, . . ..). One motivation to improve sea ice models is that this might have large consequences on atmospheric climate conditions as well.*

Author response: Yes, this is definitely an aspect that deserves to be mentioned. We added such a discussion to Introduction.

*5. P2, l2-3: See main point 1: The main focus of this study seems to be the effect on sea ice and not on ocean circulation, which is by far less intensive analyzed in this study.*

Author response: We agree, please see our answer to your Main point 1.

*6. P3, l12: I thought the minimum horizontal length is a bit smaller than 50km in the Arctic near the poles, e.g. around Greenland. Please check.*

Author response: You are correct. After checking the ORCA1 grid file, we found that the smallest grid cell lengths in the Arctic Ocean are between 40-50 km. We reworded the sentence to begin with "A typical horizontal..." from "The minimum horizontal..." as we want to tell the reader what the typical ORCA1 grid resolution is in the polar regions.

*7. P4, l17: Please explain what is meant with a salinity restoring rate of -100mm/day. If this is a freshwater flux (global average?), this sounds very large.*

Author response: The salinity restoring rate is a global negative feedback coefficient which is provided as a `namelist` parameter. The SSS restoring term should be viewed as a flux correction on freshwater fluxes to reduce the uncertainties we have on the observed freshwater budget. We added this additional information to the text. We admit that -100 mm/day is a large value. However, it is a smaller one than the default NEMO value which is -166.67 mm/day. We decided to to use the smaller value after discussions with the NEMO users of the COST EOS Ocean Synthesis action. Based on the community discussion it is likely that many NEMO users are using this, or even a higher, salinity restoring rate with ORCA1.

*8. P4, l30-P5, l2: I am confused about what tuning has been done for each of the versions? Here, it is stated that a specific and optimized tuning has been done for each of the versions. In the conclusions you state; "no specific tuning has been done". I agree that two optimized model versions should be compared. In this context, I wonder if really the same effort has been done to optimize NEMO3.6LIM2 as for optimizing NEMO3.6LIM3? My worry is that the LIM2-ice-parameters have been taken from an older NEMO-LIM2 version and that the NEMO3.6-ocean parameters have been tuned with LIM3 and not with LIM2. Please describe in more detail how these two versions have been tuned and optimized.*

Author response: This is a good comment and we think that the reviewer's concern regarding the LIM2–LIM3 comparison are justified. It is a very difficult one to address, because, in practice, there has been no systematic tuning procedure. As a result, the default parameter values of both sea-ice models are probably not the most optimal ones. They are, however, the default values obtained with the code and the ones that an average NEMO-LIM user is likely to end up using. Moreover, the systematic optimisation of both sea-ice models would have been a too daunting and complex task for this paper. Instead, we selected a more pragmatic approach and used the default parameter values. We think that this approach produces valuable results to the NEMO user community.

Regarding the detailed history of the LIM parameter values, we note that LIM2 has been used with the DFS forcing for about 10 years by the DRAKKAR community, mostly at $1/4°$ (ORCA025) resolution. The default LIM2 parameter values are a result of this exercise. Only the horizontal diffusivity (for scalability) and the EVP rheology (for numerical stability) were adjusted to the ORCA1 resolution.

The LIM3.6 default parameter values mostly come from the initial model version (Vancoppenolle et al. 2009), with some corrections on ice strength $P^*$ and albedo following Rousset et al. (2015). Both studies used a NCEP based atmospheric forcing, so it is quite comforting, and even a bit surprising, that no specific tuning of LIM3 to DFS forcing was required.

By contrast, the LIM3SC virtual sea-ice thickness parameters were specifically tuned to match two key relationships of the multi-category version: 1) the growth rate–thickness dependence, and 2) the rate of concentration decrease versus sea-ice thickness dependence.

*9. P5, l4-10: I am not sure I really understand this: Are you saying that LIM2 with P\* from LIM3 simulates much less ice volume but the same ice area than LIM2 with its standard P\*? Why is this indicating "insignificant oceanic impacts"? Please clarify.*

Author response: We have reworded the text and decided not to mention the unclear "insignificant oceanic impacts". Instead we note that the LIM2 with its standard $P^*$ results in a more realistic sea-ice volume which is why we decided to use it instead of the LIM2 simulation using the higher LIM3 $P^*$.

*10. Experiments: A table listing the different simulations would be helpful*

Author response: This is a good suggestion. We have added such a list as Table 1.

*11. P5, l15: If LIM3 with 1 ice category is much better than LIM2 but physically closer to LIM2 than LIM3, what is the reason that LIM3-1IC is better? This feature is then obviously more important for an ice model than e.g. multiple ice thickness categories.*

Author response: Suggested reasons for different performances between LIM2 and LIM3SC clearly point to differences between the thermodynamics parameterisations, including the latent heat reservoir. It is really hard to deduce the differences beyond this, because the thermodynamics code of the models are quite incompatible.

Regarding the second point, the sea-ice differences between LIM3 and LIM2, and LIM3 and LIM3SC are comparable and have the same sign. This indicates that the impact of multiple ice categories versus a single ice category is clear and systematic although when comparing LIM3 and LIM2 this is partially masked by additional LIM3–

LIM2 differences due to other differences in model configurations. The corresponding differences between LIM3SC and LIM2 are on the average smaller which signifies the primary importance of ice categories rather than the sea-ice thermodynamics parameterisations.

*12. P6, l 24: Again, I do not have the impression that this study only shortly focuses on the sea ice and merely on sensitivity experiments and oceanographic analysis. Section 3 is the longest of all sections.*

Author response: This is true, please see our answer to your Main point 1.

*13. P6, l29, Figure1 b: I do not see 50% reduction in the East-Siberian Sea. Largest reduction seems to be between North Pole and northern Kara Sea. Please check.*

Author response: Well spotted, we changed the text accordingly.

*14. Figure 1: It would be good to show the spatial distribution of LIM3MC as well.*

Author response: We added LIM3MC (now LIM3SC) spatial distributions to Figure 1.

*15. Figure2: It is not really self-explanatory to call LIM3 with 1 single ice category "LIM3MC". This sounds like LIM3 with multi-ice categories. Maybe better LIM3SC.*

Author response: True. LIM3MC stands for LIM3 mono-category, but since it may be confused with multi-category we follow your suggestion and use the LIM3SC abbreviation instead.
*16. P 7, l34: I do not think you can explain the LIM3 low summer ice extent by the ice-albedo-feedback. The ice-albedo feedback exists in reality. Maybe you can explain the low summer ice extent by too low ice albedo in LIM3 or a too strong effect due to unrealistic distribution of ice thicknesses (e.g too much thin ice). On the other hand, the annual cycle in LIM2 is even larger than in LIM3 (which is opposite to the NH). Does not this speak against an effect of the ice categories?*

Author response: Our thinking was too simplistic here. What we concluded for the spring NH sea-ice extent evolution in terms of the enhanced ice-albedo feedback due to LIM2 sub-grid-scale ice thickness distribution seem not to directly hold for the SH spring sea-ice extent evolution. We reformulated the text to explain this better:

The time series of annual mean sea-ice extent of LIM3 is rather well reproduced and closely follows observations (Figure 2d), but the sea-ice spring retreat is systematically too strong and summer extent too low. The LIM3 winter sea ice is on the average thicker than the LIM2 sea ice, while in summer their thicknesses are close to each other (not shown here). On the other hand, the average LIM3 sea-ice concentration is systematically about 1–10% smaller than the LIM2 one, even in the central ice pack. As a result, the LIM3 sea-ice extent is smaller, particularly in summer.

The processes explaining the low LIM3 summer sea-ice extent are related to (1) the steeper decline of LIM3 mean sea-ice thickness and (2) to its systematically lower sea-ice concentration. Arguably the most important process is the positive ice-albedo feedback, which is governed by the fast melting of thin ice enabling an effective penetration of solar energy into the upper ocean. Negative sea-ice–related feedbacks are the ice thickness–ice strength relationship and the ice thickness–ice

growth rate relationship which is important during the growth period. Models with sub-grid-scale ice thickness distribution have a less resistant ice pack to convergence resulting in thicker ice than a single-category model under similar conditions (Holland et al. 2006) In LIM3, this feedback exposes more open water during the melt period. In summary, the primary reason for the LIM3 low summer sea-ice extent seems to be its systematically low sea-ice concentration, and large open water fraction, which reduces the grid cell mean albedo and enhances the ice-albedo feedback. The LIM3 sub-grid-scale ice thickness distribution further enhances this feedback process, while simultaneously reducing the ice thickness–ice strength feedback.

*17. P8, l21: I think your numbers are wrong. PIOMAS shows values of about 4-8x103 km3 in September and 20-25x103 km3 in early spring in the last decade. Please check your values.*

Author response: Yes, the numbers were incorrect, thank you for pointing this out.

*18. P8, l26: LIM2 shows a stronger negative trend as LIM3? This is thus opposite to the ice area trends?*

Author response: This is the case. LIM2 has on the average thicker ice and a too small negative sea-ice extent trend. However, LIM2 has a more strongly decreasing sea-ice volume than LIM3.

*19. P9, l18: Here you state that the sea ice albedo feedback is less important in the SH. I agree but this is in contradiction to the argumentation before that the stronger sea ice albedo feedback in LIM3 explains the LIM3-low summer ice extents (see point 16).*

Author response: This contradiction does not exist any more, please see our response in point 16. Sea-ice albedo feedback occurs in NH and in SH, while it is more directly enhancing the spring sea-ice melt of LIM3 compared to LIM2 in the NH than in the SH.

*20. Figure 3: The displacement of the Beaufort Gyre in LIM3 seems to agree with the positive ice bias and the general tendency of LIM to have the thickest ice displaced/ extended towards the Northern American coast.*

Author response: This is a good observation. We have added it to the text.

*21. P10, l12/13: I do not understand: ". . .at regions." Sentence uncomplete?*

Author response: Yes, this sentence was incomplete. We changed it to: "LIM3 has a smaller ice extent and a lower ice concentration close to the ice edge (not shown here)."

*22. P 10, L25: Yes, but on the other hand LIM3MC with constant salinity does perform quite well. Without simulations that separately analyze the impact of the different new features in LIM3, it is very speculative to argue that the prognostic sea ice salinity improved ice volume and area. The results from section 4.1 do not support the conclusion that prognostic salinity is strongly improving the ice volume/ extent. Please rethink this statement here.*

Author response: We understand this and we wanted to be very careful with our wording. After rethinking we note that the impacts of the sea-ice salinity scheme appear rather small as no clear signal showing the improvements due to the prognostic

sea-ice scheme emerges from our simulations. Therefore, we decided to remove the first sentence of this paragraph.

*23. P12, l 22ff: 54 years are rather short from an ocean circulation point of view. A systematic decrease by 1-2 Sv in the AMOC might lead to larger effects on the ice after longer periods. Furthermore, I think the statement that "if problems appear, they are related to the coupling .." might be misinterpreted. In the uncoupled ocean-ice model, the atmosphere cannot feedback to the ocean. Thus, the effect of salinity/ freshwater changes on the ice is probably not very large. However, in a coupled system, small changes in the freshwater balance or related changes in AMOC and SST-pattern could lead to strong effects in the atmosphere, which in turn might strongly affect the ocean currents, ocean heat transports and ocean-ice coupling as well. Thus, changing the freshwater balance in the ocean could create important issues in a coupled model and could be the reason for performance issues in the coupled model. Please reformulate to avoid misunderstanding.*

Author response: We agree, it is likely that the differences between the simulations continue to increase if the simulations are run further. Also, it is true that in a coupled system oceanic changes modify the atmosphere which then modifies the upper ocean characteristics. We reformulated the text and do not discuss about the coupled modelling environment any more.

*24. From Fig. 5 it seems that more freshwater is transported in the East Greenland Current to the south and then further into the Labrador Sea. This might be related to the fact that there is more ice in the Greenland Sea in LIM2, which leads to a more constraint freshwater transport in the EGC in LIM2 than in LIM3. However, you relate the lower salinity in LIM2 in the Labrador Sea to more local melting. To decrease salinity this would also need a net transport of sea ice into the Labrador Sea because*

*stronger local ice growth and ice melt would not decrease the SSS. Did you analyze net ice-growth rates in LIM3 and LIM2?*

Author response: This is a good point again. We agree that it is reasonable to assume that the fresher LIM2 surface in the Labrador Sea is primarily related to the higher net transport of sea ice from the EGC, and not to the local freezing/melting of ice. We have reworded the text accordingly.

*25. Figure 5 g: I am a bit surprised over the cold bias in NEMO3.6-LIM3 in the Fram Strait-Svalbard area. Is this related to too high ice velocities and too much ice in this area? For September, Figure1 b does not seem to indicate too much ice but maybe in the rest of the year. Please add a sentence on this.*

Author response: A sentence was added. Both NEMO3.6-LIM3 and NEMO3.6-LIM2 show this cold bias (see new Figure 5i and l). We think that this is related to the DRAKKAR forcing, namely to its ERA-Interim based near-surface air temperature. Notz et al. (2013) have shown that the ERA-Interim forcing is too cold here and produces too much ice.

Notz, D., F. A. Haumann, H. Haak, J. H. Jungclaus, and J. Marotzke (2013), Arctic sea-ice evolution as modeled by Max Planck Institute for Meteorology's Earth system model, J. Adv. Model. Earth Syst., 5, doi:10.1002/jame.20016.

*26. Figures 5/6 and salinity discussion: Given the fact that SSS is quite strongly restored in NEMO-LIM2 and NEMO-LIM3, can we really conclude from the small dif-ferences in SSS that the sea ice model has a small impact on the salinity distribution? Are the SSS-differences between LIM2 and LIM3 as small in the experiments without*

*freshwater adjustments?*

Author response: To check this we plotted SSS for the experiments without freshwater adjustments and show them below and are comparable to new Figure 5d and 6d (simulations with freshwater adjustments; if plots are not visible here, please see the uploaded pdf supplement).

/tmp/1078962910/figure-1.png /tmp/1078962910/figure-2.png

It is evident that the experiments without freshwater adjustments have larger SSS differences, and the regions of significant statistical difference are larger and may have changed. For example a region north of Greenland in the Arctic Ocean is not significantly saltier in LIM3FW than in LIM2FW although that was the case when comparing LIM3 and LIM2. We have added these notes to the salinity discussion and changed the conclusion that the sea-ice model has a small impact on the surface salinity, as this is not the case in the absence of freshwater adjustments.

*27. P15, l1,2: Why is a larger Atlantic warm water inflow associated with a smaller AMOC? There is some discussion in the community how strong the AMOC is linked to the ocean heat transport into the Arctic but most studies suggest that an increased AMOC leads to increased transports of Atlantic water masses into the Arctic.*

Author response: Our statement on the link between the Atlantic warm water inflow

and AMOC is a speculative one. After reading your comment it sounds counterintutive. We decided to delete this speculative statement.

*28. Fig. 9: Are you sure that the 15% ice edge is at the right place and really the observed ice edge? It goes very far to the south and the east in the Greenland Sea and also in the Labrador Sea. Please check.*

Author response: Well spotted, thanks. The 15% ice edge is from LIM3 and not the observed one as incorrectly stated in the caption. As we want to illustrate that the mixed layer depth is shallow under sea ice, we still show the LIM3 ice edge but state it correctly in the figure caption.

*29. LIM3-Ref also indicates deep convection in the Greenland Sea far inside the ice area. Further NEMO does not show any deep convection in the Labrador Sea but the climatology does not either. Results from ARGO-floats, which cover the time period 2000-2015 (Holte et al. 2010; http://mixedlayer.ucsd.edu/) show deep convection in the Labrador Sea and might be more reliable than the climatology used in this study.*

Author response: Thank you for pointing this out and mentioning the ARGO-float based MLD estimates. We added them to Figure 9. They seem to generally agree well with the deBoyer MLDs, although differences exists. As you say, the ARGO-float based estimates show deeper MLDs in the Labrador Sea than deBoyer ones, which is now mentioned in the text.

*30. P16 AMOC: The observational based estimates should be mentioned: e.g. RAPID: 16.9 Sv (at 26.5N), Ganachaud (2003) and Lumpkin and Speer (2007): 18.5 Sv (at 24N) and 16.5 Sv at 48N (Ganachaud). There are many things well simulated*

*in NEMO, but unfortunately not the AMOC . . ..*

Author response: We agree. We added these observational estimates to the text.

*31. P17, l1-3: Again, the SSS-restoring might hide differences between LIM3 and LIM2: Is the AMOC-difference between LIM3 and LIM2 the same in the experiments without freshwater adjustments?*

Author response: To see this we added the AMOC time series of experiments without freshwater adjustments in Figure 10. For both LIM2 and LIM3, experiments without freshwater adjustments, LIM2FW and LIM3FW, have statistically significantly lower AMOCs at the 5% level than the ones with the freshwater adjustments. As the LIM3 AMOC is on the average smaller than the LIM2 AMOC, also LIM3FW AMOC is on the average smaller (0.7 Sv for 2003–2012) than the LIM2FW AMOC. The difference here is that LIM3–LIM2 AMOC difference in 2003–2012 is not statistically significant, while the LIM3FW–LIM2FW AMOC difference is (at the 5% level). It is reasonable to assume that the freshwater adjustments bring LIM3 and LIM2 AMOC closer. We mention this now in the text.

*32. P 17, l20: I would delete "briefly". As stated before: this comparison is - if not the main - but an important part of this study.*

Author response: Deleted.

*33. P17, l22 and l29-30: The conclusions on the sea ice albedo puzzle me: First you argue that the better representation of the sea ice albedo feedback is the main*

[Figure]

*improvement and then you argue that the model is stable to changes in summer albedo. The first is also related to the different thickness classes but is it really a sign for realism if the model is insensitive to the change of summer albedo? How much is the summer albedo changing with the new sea ice albedo scheme? Maybe, the difference is small?*

Author response: The other reviewer was also puzzled. Our wording is misleading and provides a view that overestimates the effect of the new sea-ice albedo scheme. The LIM3 sea-ice thickness categories enhances the ice-albedo feedback than the single-category LIM2 which has been shown by Holland et al. (2006), for example, and discussed earlier. The new albedo scheme provides better transitions between the different ice types, slightly modifies the surface albedo compared to the old scheme and affects the model behaviour to a limited extent only. Therefore, the impact of the new sea-ice albedo scheme is secondary compared to the sea-ice thickness categories. We now mention this in Conclusions.

*34. P 18, l3: I think it is a bit overstated to say that you evaluated the oceanic transports across major transects of the world ocean. You only looked at the AMOC and the Drake Passage. You do not show any results from ocean heat transports in the different oceans or transports into the Arctic or overflows (Denmark Strait, IcelandFaroe-Scotland).*

Author response: In the manuscript, we only show the AMOC at 50–53°N and the Drake Passage volume transports, but we have calculated and compared other transports (volume, heat and salinity) as well. We decided not to include plots of these other transports in the manuscript due to their similarities between LIM2 and LIM3 or because they did not provide any important additional information, and for not to increase the number of figures too high and not to extend the manuscript too
long. Specifically, we calculated the oceanic transports for AMOC across 20–23°N, 30–33°N, 40–43°N, 45–48°N and 50–53°N. Moreover, we calculated time series across the Australia–Antarctica transect, the Bering Strait, the Denmark Strait, the Drake Passage, the Florida Strait, the Gibraltar Strait, and the Greenland–Norway transect at 60°N. This has now been mentioned in the text.

**Typings, etc.**

*P1, l7"while NEMO-LIM2 deviates more". Could be deleted, if LIM3 agrees better than it is clear that LIM2 deviates more.*

Author response: Deleted.

*P13, l6/ l7: "melted freshwater" sounds weird. Better: "freshwater from melted sea ice"*

Author response: Changed to "freshwater melted from excessive sea ice."

*P 14, l10: delete one "be"*

Author response: Deleted.

*P17, l13: A set . . . "was" performed.*

Author response: Corrected.

[Figure]

*P 17, l13: ". . . in the global ORCA1 grid": Add a "configuration" or "using the global ORCA1 grid.*

Author response: Added.

---

## Author Response (AR2)

**Author's response**

Please find our responses the Reviewer and Editor requests below. In general, we have changed the manuscript accordingly. Following the Reviewer comment on Mixed Layer Depth, we rewrote parts of the section in question.

**Reviewer comments**

*Page 1, Line 13: This last sentence of the abstract sounds a bit strange. At least the "in" after "effective"should be deleted but I would suggest to reformulate the entire sentence. Do you mean that the drift is due to a stronger deep water formation around Antarctica in LIM3?*
Response: Thank you for your comment. We agree that the last sentence sounds strange and we rewrote it. We also did some minor edits in the abstract to improve its clarity and readability. You are right, the drift is probably due to a somewhat stronger deep water formation.

*Page 4, Line 10: ETOPO1 (Amante et al., 2009)*
Response: Corrected.

*Page 8, Line 18: "modeli"*
Response: Corrected.

*Page 9, line 18: I think, the ice thickness should be the ratio between ice volume and ice area and not ice extent (otherwise ice free areas are also counted for ice concentrations > 15%).*
Response: This is true, we have changed the text in the paragraph accordingly.

*Page 9, lines 27-31: If this is explaining the difference in trends in GIOMAS and LIM2/ LIM3 in the SH, maybe you should call it "GIOMAS" instead of "PIOMAS".*
Response: Well spotted, thanks. We are actually discussing reasons for differences in both hemispheres. Due to this we changed PIOMAS to P/GIOMAS.

*Page 11, line 1: ... faster than LIM2 ice motion.*
Response: Corrected.

*Page 11, Sea ice salinity: It seems realistic that LIM3 simulates lower ice salinities in summer than in winter. Are there any observations of ice salinity available? If not, it might be worth to state this in the text.*
Response: There are salinity observations detailed in Vancoppenolle et al. (2009b). They are consistent with the simulated patterns. We now mention this in the text.

*Page 13, line 1: delete one "PHC3"*
Response: Deleted.

*Page 13, line 26 ... surface is saltier...*
Response: Changed accordingly.

*Page 16, Mixed Layer Depth: Mixed layer in NEMO might be deeper than in climatologies in many regions but are they really deeper in the deep water formation areas in Labrador Sea and Greenland Sea? It is hard/ impossible to see in the figure 9 because the figures are small but 9a) shows that NEMO-LIM3 has a mean March MLD of below 500m. If I look at the ARGO-climatology (Holte et al. 2016), I find March-mean values of up to 1000m in the Labrador Sea. To*

*my understanding, the weak deep water formation, particularly in the Labrador Sea, in NEMO is the main reason for the weak AMOC in NEMO.*

Response: We agree with your reasoning, and as a result we double checked our routine used to plot Figure 9. We found an error in the routine: instead of subtracting monthly, March in the NH and September in the SH, climatology fields from NEMO MLDs, we had subtracted their annual means. After correcting this error the climatology MLDs used in Figure 9 became deeper and our findings now agree very well with the referee comment. We have rewritten the related text in section 5.9 "Mixed Layer Depth".

It is true that the shallower than ARGO-observed LIM MLDs in the Greenland Sea are difficult to see in Figure 9c and 9f due to their small size. One way to ease their detection is to enlarge the figure panels on a computer screen. This might be an adequate approach as science publications are typically read digitally nowadays.

**Editor comments:**

*The title is improved from the last version, but I do not like the word "some", and also the order of the things you are comparing. The first thing you look at is sea ice, and it is the bulk of the paper, so this should come first in the list. Please revise.*

Response: We dropped the word "some"and changed the order of things as suggested.

*I do not think the change to the colours in Fig.3 will improve the situation in relation to the issue of color-blindness, I suggest using different colours away from red and green, rather than just changing the tone as you have done.*

Response: You are right. We have changed the colour of the hatching in Figure 3 from green to turquoise.

*The paper has a lot of acronymns, particularly on page 6, many of which are undefined, while some are common and may be assumed, I think in particular OPA on page 3, and PHC3 on page 6 should be spelt out.*

Response: A good point, thanks. We have now spelt most acronyms out.

*page 16: line 12, should be "LIM3 becomes, on average, saltier"*
Response: Changed accordingly.